# Optical and chemical properties and oxidative potential of aqueous-phase products from OH and $^3C^*$-initiated photooxidation of eugenol

Xudong Li[1], Ye Tao[1], Longwei Zhu[1], Shuaishuai Ma[1], Shipeng Luo[1], Zhuzi Zhao[1], Ning Sun[1], Xinlei Ge[2,*], Zhaolian Ye[1,*]

[1]College of Chemistry and Environmental Engineering, Jiangsu University of Technology, Changzhou 213001, China

[2]Jiangsu Key Laboratory of Atmospheric Environment Monitoring and Pollution Control, Collaborative Innovation Center of Atmospheric Environment and Equipment Technology, School of Environmental Sciences and Engineering, Nanjing University of Information Science and Technology, Nanjing 210044, China

*Correspondence: Zhaolian Ye (bess_ye@jsut.edu.cn) and Xinlei Ge (caxinra@163.com)

**Abstract:** Aqueous reactions may turn precursors into light-absorbing and toxic products, leading to air quality deterioration and adverse health effects. In this study, we investigated comprehensively eugenol photooxidation (a representative biomass burning emitted, highly substituted phenolic compound) in bulk aqueous phase with direct photolysis, hydroxyl radical (OH) and an organic triplet excited state ($^3C^*$). Results show that the degradation rates of eugenol followed the order of $^3C^*$>OH >direct photolysis. During $^3C^*$-initiated oxidation, different reactive oxygen species (ROS) including $^3C^{*,}$ OH, $^1O_2$ and $O_2^{\bullet-}$ can participate in oxidation of eugenol, quenching experiments verified $^3C^*$ was the most important one; while during OH-initiated oxidation, $O_2^{\bullet-}$ was a more important ROS than OH to degrade eugenol. The rate constants under saturated $O_2$, air and $N_2$ followed the order of $k_{O_2} > k_{Air} > k_{N_2}$ for

both direct photolysis and OH-initiated oxidation, but changed to $k_{Air} > k_{N_2} > k_{O_2}$ for $^3C^*$-mediated oxidation. pH and dissolved oxygen (DO) levels both decreased during oxidation, indicating formation of acids and the participation of DO in oxidation. UV-vis light absorption spectra of the reaction products showed clear absorbance enhancement in the 300-400 nm range for all three sets of experiments and new fluorescence at excitation/emission=250/(400-500) nm appeared, suggesting the formation of new chromophores and fluorophores (brown carbon species); and these species were likely attributed to humic-like substances (HULIS) as shown by the increases of HULIS concentrations during oxidation. Large mass yields of products (140%-197%) after 23 hours of illumination were obtained, and high oxidation degrees of these products were also observed; correspondingly, a series of oxygenated compounds were identified, and detailed reaction mechanism with functionalization as a dominant pathway was proposed. At last, dithiothreitol (DTT) assay was applied to assess oxidation potential of the reaction products, and the end products of all three sets of experiments showed higher DDT consumption rates than that of eugenol, indicating more toxic species were produced upon aqueous oxidation. Overall, our results by using eugenol as a model compound, underscore the potential importance of aqueous processing of biomass burning emissions in secondary organic aerosol (SOA) formation.

## 1 Introduction

Photochemical reactions in atmospheric aqueous phases (cloud/fog droplets and aerosol water) can affect lifetimes of many organic species, and are an important source and pathway of secondary organic aerosol (SOA) formation (Vione et al., 2006; Zhao

et al., 2012). Compared to the SOA formed via gas-phase photochemical oxidation
(gasSOA), aqueous-phase SOA (aqSOA) is often more oxidized and less volatile,
therefore might play an important role in haze formation, air quality and global climate
change (Ervens et al., 2011; Lim et al., 2010). However, due to complexity of the
aqueous reactions and influencing factors (such as precursors, oxidants, and light
intensities), detailed reaction mechanism, optical property, oxidative potential (OP) and
the interplay among them remain poorly understood.
Many laboratory studies have focused on aqueous-phase oxidations of low
molecular weight (LMW) volatile organic compounds (VOCs), such as isoprene,
terpenes (α-, β-pinene), as well as their gas-phase oxidation products (such as glyoxal,
methylglyoxal, *cis*-pinonic acid and methyl vinyl ketone) (Faust et al., 2017; Herrmann,
2003; Herrmann et al., 2015; Huang et al., 2011; Lee et al., 2012; Zhang et al., 2010).
Recently, aqueous oxidation of semi-/intermediate volatility VOCs (S/IVOCs), such as
the phenolic compounds emitted from combustion or pyrolysis of lignin in biomass,
were also extensively investigated (Barzaghi and Herrmann, 2002; Bonin et al., 2007;
Chen et al., 2020; Gilardoni et al., 2016; He et al., 2019; Jiang et al., 2021; Li et al.,
2014; Li et al, 2021; Ma et al., 2021; Mabato et al., 2022; Smith et al., 2014; Sun et al.,
2010; Tang et al., 2020; Yang et al., 2021; Yu et al., 2016; Lu et al., 2019). Generally,
chemical structures of precursors have profound influences on the reaction mechanisms
and products, while effect of oxidants also cannot be neglected. It is evident that liquid
water can contain various types of oxidants, such as singlet oxygen ($^1O_2$), nitrate radical
($NO_3$), hydroxyl radical (OH), and organic triplet excited states ($^3C^*$), and all oxidants
can play crucial roles in photooxidation reactions (Kaur and Anastasio, 2018; Scharko
et al., 2014). Among them, OH is a ubiquitous oxidant with concentrations of $10^{-13}$-$10^{-12}$
$^{12}$ mol·$L^{-1}$ (Arakaki et al., 2013; Gligorovski et al., 2015; Herrmann et al., 2003). Hence,
aqueous OH-induced photooxidation has been extensively studied (Chen et al., 2020;
Sun et al., 2010; Yu et al., 2016). Compared to OH oxidation, $^3C^*$-initiated aqueous
oxidation (photosensitized reactions) has also attracted attentions in recent years (Ma et
al., 2021; Wang et al., 2021). Several classes of organic compounds in ambient air,
including non-phenolic aromatic carbonyls, quinones, aromatic ketones and nitrogen-
containing heterocyclic compounds, can form $^3C^*$ after absorbing light (Alegría et al.,
1999; Kaur et al., 2019; Nau and Scaiano, 1996; Rossignol et al., 2014; Chen et al.,
2018). These compounds are termed photosensitizers. $^3C^*$ is capable of reacting with
$O_2$ to produce singlet oxygen ($^1O_2$) and superoxide radicals ($O_2^{\cdot-}$). Various reactive
oxygen species (ROS) can be generated and affect greatly the $^3C^*$-initiated aqueous-
phase reactions. Despite some studies demonstrating importance of ROS in
photochemical process (Ma et al, 2021; Wang et al., 2020; Wang et al., 2021), our
current understanding on $^3C^*$-initiated oxidation is still limited.

Excitation-emission matrix (EEM) fluorescence spectroscopy, as a low-cost, rapid,

non-destructive and high-sensitivity technique, can offer detailed information on
chromophores hence has been widely employed for studies of aquatic dissolved organic
matter (Aryal et al., 2015). Nevertheless, it has not been extensively used in
atmospheric aerosol research (Mladenov et al., 2011). Prior studies have investigated
the relationship between the fluorescence components and chemical structures of
atmospheric aerosols by using high-resolution aerosol mass spectrometry (AMS) and
EEM fluorescence spectroscopy (Chen et al., 2016a; Chen et al., 2016b). An earlier
report from Chang and Thompson (2010) found fluorescence spectra of aqueous-phase
reaction products of phenolic compounds, which had some similarities with those of
humic-like substances (HULIS), and Tang et al. (2020) reported that aqueous
photooxidation of vanillic acid could be a potential source of HULIS. Chang and

Thompson (2010) also showed that light-absorbing and fluorescent substances generally had large conjugated moieties (i.e., quinones, HULIS, polycyclic aromatic hydrocarbons (PAHs)), which can damage human body (Dou et al., 2015; McWhinney et al., 2013). HULIS are considered as an important contributor to induce oxidative stress since they can serve as electron carriers to catalyze ROS formation (Dou et al., 2015; Ma et al., 2019; Huo et al., 2021; Xu et al., 2020), causing adverse health effects. Dithiothreitol (DTT) assay (Alam et al., 2013; Verma et al., 2015), as a non-cellular method, was widely employed to determine oxidation activity and OP of atmospheric PM (Chen et al., 2019; Cho et al., 2005) for the evaluation of its health effects. Some other works (Fang et al., 2016; McWhinney et al., 2013; Verma et al., 2015; Zhang et al., 2022) focused on the link between chemical components and OP in PM, and confirmed that several kinds of compounds, such as quinones, HULIS and transition metals usually had strong DTT activities. However, DTT method is rarely used to evaluate the OP of aqueous-phase oxidation products previously (Ou et al., 2021).

In the present work, we chose eugenol (ally guaiacol) as a model compound to conduct aqueous oxidation experiment. As a representative methoxyphenol emitted from biomass burning (BB) (Hawthorne et al., 1989; Simpson et al., 2005), it was widely detected in atmospheric particles. For instance, concentration and emission factor of this compound from beech wood burning were 0.032 μg/m$^3$ and 1.534 μg/g, which were twice those of guaiacol (0.016 μg/m$^3$ and 0.762 μg/g) (Bari et al., 2009). Eugenol is a semivolatile aromatic compound with a moderate water-solubility (2.46 g/L at 298 K). Chemical characteristics of aqueous reaction products under direct photolysis (without oxidant) and oxidations by OH radicals and $^3$C* triplet states, were comprehensively elucidated by a suite of analytical techniques including high-performance liquid chromatography (HPLC), ultraviolet and visible (UV-Vis)

spectrophotometry, gas chromatography mass spectrometry (GC-MS), and soot particle

aerosol mass spectrometry (SP-AMS). Moreover, light absorption, fluorescent and

oxidative properties of the aqueous oxidation products were also investigated.

## 2 Materials and methods

### 2.1 Chemicals and reagents

Eugenol (99%), tert-butanol (TBA, 99%), 3,4-dimethoxybenzaldehyde (DMB,

99%), para-benzoquinone ($p$-BQ, 99%), dithiothreitol (99%) and 5,5′-dithiobis-2-

nitrobenzoic acid (DTNB, 99%), 2-nitro-5-thiobenzoic (99%), 5,5-dimethyl-1-

pyrroline N-oxide (DMPO)**,** 2,2,6,6-tetramethylpiperidine (TEMP) were all purchased

from Sigma-Aldrich. Superoxide dismutase (SOD) was purchased from Bovine

Erythrocytes BioChemika. Dichloromethane (HPLC-MS grade, 99%), methanol

(HPLC-MS grade, 99%), acetonitrile (HPLC-MS grade, 98%), hydrogen peroxide

($H_2O_2$, 35 wt %), and 2,4,6-trimethylphenol (TMP, 99%) were all obtained from Acros

Chemicals. Sodium azide ($NaN_3$, 98%) was purchased from J&K Scientific Ltd.

(Beijing, China). All solutions were prepared using ultrapure water (Millipore) on the

days of experiments.

### 2.2 Photochemical oxidation experiments

Aqueous-phase photochemical reactions were carried out in a Rayonet

photoreactor (model RPR-200) equipped with 16 light tubes (2 RPR-3000, 7 RPR-3500

and 7 RPR-4190 tubes), which was frequently used to mimic sunlight for

photochemical experiments and was described in details by several groups (George et

al., 2015; Hong et al., 2015; Huang et al., 2018; Jiang et al., 2021; Zhao et al., 2014).

Pyrex tubes containing sample solutions were placed in the center and received
radiation from surrounded lamps of all sides. To ensure mixing of the solution, a fan
and a magnetic stir bar were placed at the bottom of the reaction tube. The solution
temperature was controlled at $25\pm2°C$. The same photoreactor system and a normalized
distribution of photon fluxes inside the reactor have been reported elsewhere (George
et al., 2015), and the wavelength of light was in the range of 280~500 nm. We only
measured light intensity at the surface of the solution with a radiometer (Photoelectric
instrument factory of Everfine Corporation, Hangzhou, China), which was determined
to be ~2400 $\mu W/cm^2$ in the range of 290-320 nm (UVB), lower than the sunlight
intensity (6257.1 $\mu W/cm^2$).
In this work, 300 $\mu M$ $H_2O_2$ and 15 $\mu M$ DMB were added into solutions as sources
of OH and $^3C^*$, respectively. The initial concentration of eugenol was 300 $\mu M$. For $^3C^*$-
mediated experiments, solutions were adjusted to pH=3 by sulfuric acid in order to
perform experiments under optimal conditions (Ma et al., 2021; Smith et al., 2014)
since DMB triplet state is protonated to a more reactive form in acidic solution. We
conducted three sets of oxidation experiments: (A) 300 $\mu M$ eugenol + 300 $\mu M$ $H_2O_2$,
(B) 300 $\mu M$ eugenol + 15 $\mu M$ DMB, and (C) 300 $\mu M$ eugenol without oxidants. In each
series of experiments, a dark control experiment was performed synchronously with a
Pyrex tube wrapped by aluminum foil. Results showed loss of eugenol under dark
conditions were negligible (data not shown). In addition, to evaluate the roles of ROS
in eugenol degradation during $^3C^*$-initiated oxidation, quenching experiments by using
specific scavengers to capture different ROS were performed, namely TBA for OH,
$NaN_3$ for $^1O_2$, SOD for $O_2^{\bullet-}$, and TMP for $^3C^*$, respectively (Pan et al., 2020; Chen et
al., 2020). For OH-initiated oxidation, quenching experiments using $p$-BQ for $O_2^{\bullet-}$ (Ma
et al., 2019; Raja et al., 2005), and TBA for OH were conducted. For most experiments,
solutions were saturated by air and each experiment presented was repeated three times
unless otherwise stated. Average results with one standard deviation were provided. In
order to further evaluate the role of oxygen in photooxidation, experiments were also
conducted by using different saturated gases (air, $N_2$ and $O_2$).
**2.3 Analytical methods**
**2.3.1 Determination of eugenol concentrations**
Before and during the photochemical experiment, 2 mL of reacted solution was
sampled periodically and subjected to HPLC (LC-10AT, Shimadzu, Japan) analysis to
quantify eugenol concentration. The HPLC was equipped with an InertSustain AQ-C18
reverse phase column (4.6×250 mm, 5.0 μm, Shimadzu) and a UV-vis detector. The
mobile phase was a mixture of acetonitrile/$H_2O$ (v/v: 60/40) at a flow rate of 0.6
mL/min, and the detection wavelength was 280 nm. The first-order kinetic rate constant
of eugenol degradation can be obtained from the slope of plot of -ln($c_t$/$c_0$) versus
reaction time, as presented in Eq.(1).

$$\ln(c_t/c_0)=-kt \qquad\qquad (1)$$

Where $c_0$ and $c_t$ are eugenol concentrations (in μM) at the initial and reaction time
t, while k represents the pseudo first-order rate constant (in $s^{-1}$).
**2.3.2 UV-vis and fluorescent spectra**
The UV-vis light absorption spectra of reacted solutions (placed in a 1 cm path
length quartz cuvette) were measured by using an UV-vis spectrophotometer (Specord
plus, Analytik Jena., Germany). The instrument has a dual-beam optical system
with tungsten and deuterium lamps as light sources. A reference absorption spectrum

of ultrapure water was carried out in the same cuvette prior to sample analysis for baseline correction.

Immediately after the UV-Vis measurement, the cuvette was transferred to a three-dimensional EEM fluorescence spectrometer (FluoroMax Plus,HORIBA Scientific). The ranges of wavelength varied from 200 to 450 nm for excitation wavelengths (Ex) and from 290 to 650 nm for emission wavelength (Em). Intervals of the excitation and emission wavelengths were 5 nm and 2 nm, respectively. The reported absorbance and EEM spectra here are averages of the results from experiments in triplicate.

**2.3.3 Determination of HULIS concentrations**

Solid phase extraction (SPE) cartridges (CNW Poly-Sery HLB, 60 mg/cartridge) were used to isolate HULIS from the reaction products. The SPE cartridge was first rinsed with 1 mL ultrapure water and 3 mL methanol prior to extraction. The solution was acidified to pH ~2 using HCl and loaded on an SPE cartridge, which was rinsed with 1 mL ultrapure water again. Next, 3 mL methanol/ammonia (98:2, v/v) mixture was added into the SPE cartridge to elute HULIS, and the solution was blown to full dryness with high purity $N_2$, followed by dilution with ultrapure water to 25 mL for quantification of HULIS using the HPLC coupled with an evaporative light scattering detector (ELSD3000). Recovery efficiency of the HULIS standard, Suwanne River Fulvic Acid (SRFA), was 75-80% with the standard deviation of reproducibility less than 5%. More details have been described elsewhere (Tao et al., 2021).

**2.3.4 Oxidative potential (OP) based on DTT assay**

The OP of reaction products was determined by the DTT method (Cho et al., 2005; Lin and Yu, 2019) with slight improvements. Briefly, 1.2 mL sample solution was

transferred into a 10 mL glass tube, then 6 mL phosphate buffer (0.1 M, pH 7.4) and

300 μL of 2.5 mM DTT were added and mixed thoroughly. The DTT mixed solution

was placed in a 37°C water bath for incubation. Over the course of reactions that lasted

for 150 minutes, 1 mL aliquot of DTT mixture was taken every 30 minutes, and 100 μL

of 5 mM DTNB (prepared in 0.1 mM phosphate buffer) was added and loaded in a

centrifuge tube. Next, reactions between DTNB and DTT produced bright yellow TNB,

which was quantified by the UV-Vis spectrometer within 30 minutes. Finally, we

measured the light absorbance ($A_t$) at 412 nm to indirectly quantify the remaining DTT.

Another 1.2 mL ultrapure water instead of sample solution was treated in the same way

and the absorbance was denoted as A as the blank value. $A_0$ represents the initial light

absorbance value. Thus, DTT concentration consumed by the sample solution ($M_{DTT}$,

μM) and that by the blank solution ($M_{DTT0}$, μM) can be calculated according to Eq.(2)

and Eq.(3), respectively.

$$M_{DTT} = \frac{A_0 - A_t}{A_0} \times C_{DTT_0} \tag{2}$$

$$M_{DTT0} = \frac{A_0 - A}{A_0} \times C_{DTT_0} \tag{3}$$

Here, $C_{DTT0}$ was the initial DTT concentration in sample solution (100 μM in this

work). DTT consumption rates ($R_{DTT}$ and $R_{DTT0}$) were then obtained from the slopes of

plots of $M_{DTT}$ and $M_{DTT0}$ versus incubation times. Experiments of blanks and samples

were typically run in a triplicate. The reproducibility of the whole analysis showed that

the relative standard deviation of DTT consumption rate was 3-4%.

**2.3.5 Product analysis by GC-MS**

Reacted solution (about 30 mL) was extracted with 10 mL dichloromethane twice.

The extract was concentrated into 1 mL by blowing $N_2$ gently, subsequently transferred

to a 2 mL vial, and analyzed by a GC-MS (7890A GC/5975C MS, Agilent) with a DB-

5ms capillary column (30 m×0.25 mm×0.5 μm). The operational conditions were set as
follows: injector was at 200℃; ion source was at 230 ℃; column oven temperature was
programmed to be held at 35℃ for 4 minutes, then ramped to 250 ℃ at a rate of
20℃/minute and held for 10 minutes. The recovery efficiency, method detection limits
and quality assurance/quality control have been described in our previous work (Ye et
al., 2020).

**2.3.6 SP-AMS analysis and mass yields of reaction products**

An Aerodyne SP-AMS (Onasch et al., 2012) was applied to analyze the low-
volatility organic products, similar to our previous work (Chen et al., 2020; Ge et al.,
2017). SP-AMS data were acquired in V mode and analyzed by Squirrel v.1.56D and
Pika v1.15D software. The organic fragments were classified into six groups: CH, CHO,
CHN, $CHO_2$, CHON and HO. Elemental ratios (oxygen-to-carbon, O/C; hydrogen-
to-carbon, H/C), were calculated according to the method proposed by Canagaratna et
al. (2015).
Since the AMS analysis requires nebulization of sample solution into particles
before determination, and quantification of organics was influenced by the atomization
efficiency and carrier gas flow, we thus cannot use SP-AMS measured concentration to
quantify the mass of products directly. In this case, according to Li et al. (2014), we
added an internal standard ($SO_4^{2-}$) prior to AMS analysis, and the mass ratio of particle-
phase organics to $SO_4^{2-}$ ($\Delta Org/SO_4^{2-}$) can be used to calculate the mass concentration
of products. Furthermore, the mass yield of aqueous-oxidation products ($Y_{products}$, %),
which is the mass of products generated per unit mass of precursor consumed, can be
calculated according to Eq. (4).

$$Y_{products}(\%) = \frac{(\Delta Org/SO_4^{2-})[SO_4^{2-}]_0}{c_0 M \eta} \times 100\% \qquad (4)$$

Where $[SO_4^{2-}]_0$ is the $SO_4^{2-}$ concentration (here 7.27 mg/L), $C_0$ is the initial eugenol
concentration (in mmol/L), M is MW of the precursor (164 g/mol for eugenol) , and η
is the degraded fraction of eugenol.

## 3 Results and discussion

### 3.1 Kinetics of aqueous photooxidation

Figure 1 shows unreacted eugenol concentrations ($c_t$) and the negative logarithm
of $c_t/c_0$ (-ln($c_t/c_0$)) as a function of reaction time, respectively. The pseudo first-order
rate constants (k) obtained by Eq.(1) were also presented. As described in Fig. 1a,
eugenol concentration decreased to be <20% of the initial concentration in 3 hours,
suggesting photooxidation was fast under all three reaction conditions. In the presence
of $^3C^*$, eugenol was degraded nearly 100% after 3 hours. Previous study (Chen et al.,
2020) on $^3C^*$-initiated 4-ethylguaiacol oxidation reports a time of 21 hours for a
complete degradation. Apart from difference of precursors, different light irradiation
spectra and stronger energy of light in this work than the previous work might be
responsible for the fast loss of eugenol. The bond dissociation energies (BDEs) are 340
kJ/mol for OH, 374 kJ/mol for C-H in $-CH_3$ group, 345 kJ/mol for C=C bond, and 403
kJ/mol for C-H in $-OCH_3$ group, respectively (Herrmann et al., 2003; He et al., 2019).
Due to influences of steric hindrance and intramolecular hydrogen bonding, the H-
abstraction from OH group might not be favorable and the most probable H-abstraction
might take place in C=C of the allyl group. As a result, breakage of C=C into C-C at
the allyl group can lead to the formation of 2-methoxy-4-propyl-phenol (Section 3.6.1).
When photon energy is higher than the BDE, chemical bonds can break, leading to
decomposition of compounds and possibly further mineralization. The energy of photon

of 300 nm is 412 kJ/mol and can break all major bonds in eugenol, while the energy of

350 nm is 353 kJ/mol, being able to break some of the bonds in eugenol as well. Overall,

eugenol can be easily decomposed after absorbing the photons.

As shown in Fig. 1b, the first-order rate constants were $2.43\times10^{-4}$ $s^{-1}$, $2.73\times10^{-4}$ $s^{-1}$, and $5.75\times10^{-4}$ $s^{-1}$ for direct photolysis and photooxidations by OH and $^3C^*$, respectively. $^3C^*$-initiated photooxidation was quicker than that attacked by OH, likely due to combined contributions from reactions with $^1O_2$, $O_2^{\bullet-}$ and OH (Section 3.2). Similar results were found for aqueous phase reactions of three phenols against OH and $^3C^*$ by Yu et al. (2016) (Note the initial concentrations of $H_2O_2$ and DMB were 100 μM and 5 μM, respectively, with the same ratio as 300 μM $H_2O_2$ to 15 μM DMB in this work)

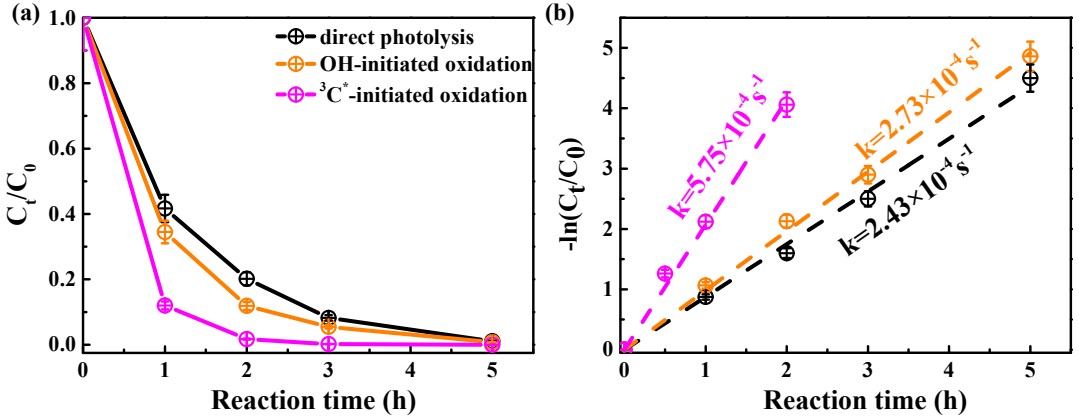

**Figure 1.** Aqueous-phase eugenol decay kinetic curves (a) and first-order rate constants (b) obtained

based on Equation 1 under direct photolysis, OH-initiated oxidation and $^3C^*$-initiated oxidation.

Error bar represents one standard deviation from the measurements in triplicate.

## 3.2 Relative importance of ROS in photooxidation

### 3.2.1 Quenching experiments in $^3C^*$-initiated photooxidation

Relative importance of different ROS in photooxidation can be investigated by

addition of scavengers/quenchers, and then be evaluated based on the different

degradation efficiencies of eugenol in absence and presence of the corresponding ROS
quenchers. For each quencher, we conducted several gradient experiments with varying
molar ratios of eugenol to quencher. The ratios were 0.075:1, 0.15:1, 0.3:1, 0.75:1, 1.5:1
for quenchers of $NaN_3$, TMP and TBA, and 1.2:1, 1.6:1, 2.5:1, 5:1, 10:1 for SOD, which
were all within the typical ranges of ROS quenching experiments reported previously
(Zhou et al., 2018). Excess concentrations of quenchers have been added repeatedly to
ensure the complete reactions between ROS and scavengers. Figure 2 displays the
impacts of quenchers on eugenol degradation. All rate constants (k) with quenchers
were lower than those of the quencher-free solutions. The optimum molar ratio of
eugenol to quencher was selected such that the eugenol degradation did not change with
the increase of added quencher (Wang et al., 2021). For example, along with the
decrease of molar ratios of eugenol to $NaN_3$ from 1.5:1 to 0.075:1, the variation of
eugenol degradation was stabilized at the ratio of 0.15:1, indicating that $^1O_2$ has been
completely quenched at this ratio, therefore a molar ratio of 0.15:1 for $NaN_3$ was
optimal, since excess scavenger may generate other products that interfere the existing
reactions. Finally, the optimal molar ratios of eugenol to quenchers of TBA, $NaN_3$, TMP
and SOD, were determined to be 1.5, 0.15, 0.075 and 2.5, respectively. Table 1 and Fig.
S1 compared the rate constants determined under the ratios above and they were in an
order of TMP<$NaN_3$<SOD<TBA, suggesting relative importance of generated ROS to
eugenol degradation was in the order of $^3C^*$＞$^1O_2$＞$O_2^{\cdot-}$＞OH. This result suggests that
$^3C^*$ plays a major role among all ROS in the photooxidation. Previously, Laurentiis et
al. (2013) reported that 4-carboxybenzophenone (70 μM) could act as $^3C^*$ and the
photosensitized degradation was more effective than oxidants such as OH, $O_3$;
Misovich et al. (2021) investigated the aqueous DMB-photosensitized reaction (5 μM,
same as in this study) also demonstrated that $^3C^*$ was the greatest contributor to phenol

or guaiacyl acetone degradation, followed by $^1O_2$, while both OH and $^1O_2$ contributions were relatively minor.

To further assess the relative importance of different ROS, we propose to use the following Eq.(5) for a rough estimation:

$$Red_{ROS} \text{ (in \%)} = (k - k_{quencher})/k \times 100\% \qquad (5)$$

Here k (in $s^{-1}$) is the original rate constant of $^3C^*$-initiated oxidation (or OH-initiated oxidation in Section 3.2.2) and $k_{quencher}$ (in $s^{-1}$) is the rate constant after the target ROS has been completely scavenged by its quencher. k and $k_{quencher}$ in fact refer to those reported in Fig. S1b. $Red_{ROS}$ then refers to the percentages of reduction due to addition of quencher for a ROS.

According to Eq.(5), $Red_{3C^*}$ was calculated to be 85.7%. Similarly, the values of $^1O_2$, $O_2^{\cdot-}$ and OH were 80.5%, 61.4% and 53.9%, respectively. Note $Red_{ROS}$ only reflects the relative important of ROS and it does not corresponds to the exact contribution of that ROS in eugenol degradation without quenchers. The reason is that although the addition of a ROS scavenger can eliminate oxidation by this ROS, but it also significantly interrupts the original chain reactions as compared to those in the absence of the scavenger, and reactions with other ROS might be enhanced. In this regard, the sum of the four $Red_{ROS}$ values may exceed 100%. Therefore, one should be cautious to use $Red_{ROS}$ as a precise quantification of the ROS contribution in aqueous oxidation. Determination of ROS concentrations during oxidation should be instead be an effective way to elucidate the role of ROS. Here, we tried to detect in-situ generated OH, $O_2^{\cdot-}$ and $^1O_2$ during photochemical reactions using a micro electron spin resonance (ESR) spectrometer (Bruker Magnettech, Berlin, Germany) with DMPO as the spin trap to form stable DMPO-OH or DMPO-$O_2^{\cdot-}$, with TEMP to capture $^1O_2$ to produce TEMP-$^1O_2$ spin-adduct (TEMPO). The radicals can be identified and quantified by the peak

patterns in ESR spectra, such as the quarter line with a height ratio of 1:2:2:1 for
DMPO-OH, 1:1:1:1 for DMPO-$O_2^{\cdot-}$ and 1:1:1 for TEMP-$^1O_2$ (Guo et al., 2021).
Unfortunately, OH radical cannot be detected since its concentration might be lower
than the detection limit of the instrument (Fig. S2, ESR spectra of OH). In contrast, we
were able to detect high concentrations of $^3C^*$ and found the intensity of TEMP-$^1O_2$
signal reached its maximum at 30 minutes, then decreased slowly (Fig. S2, ESR spectra
of $^1O_2$). Combining the great quenching effect of TMP with high $^1O_2$ concentration from
ESR method, we can conclude that $^3C^*$ and $^1O_2$ play relatively important roles in
eugenol photooxidation.

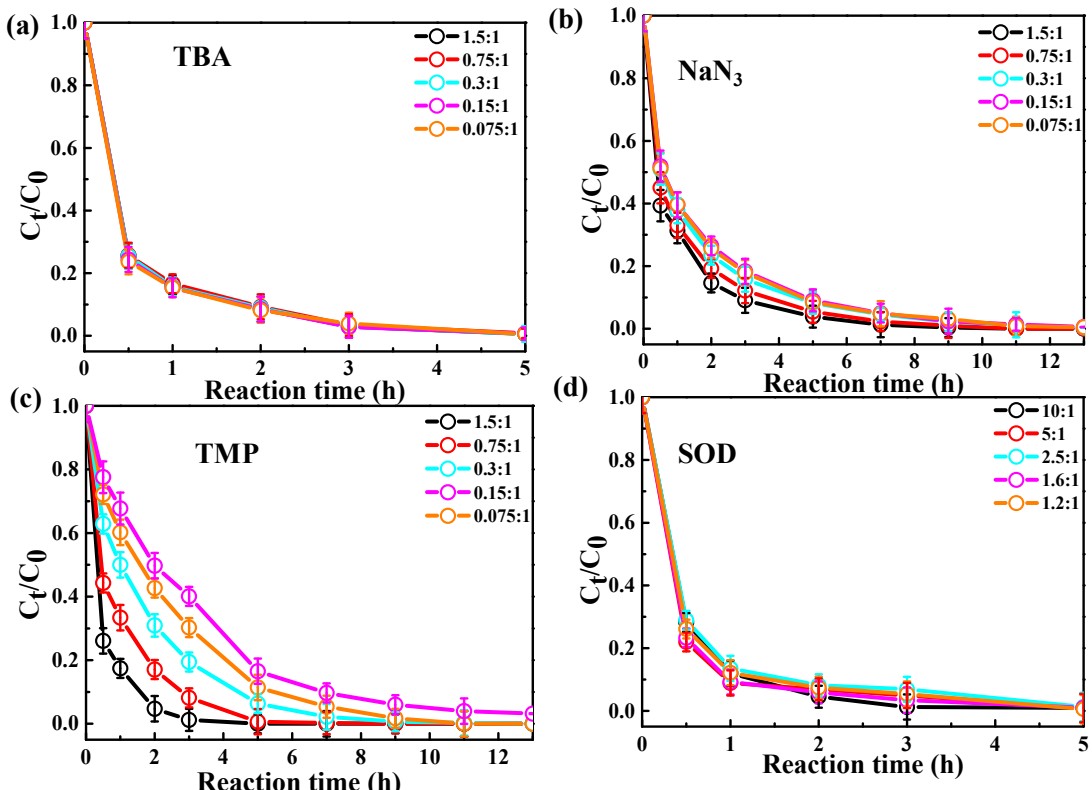


**Figure 2.** Ratio of unreacted eugenol concentration to its initial concentration ($C_t/C_0$) at different
molar ratios of eugenol to quencher, as a function of reaction time: (a) TBA, (b) NaN$_3$, (c) TMP and
(d) SOD.

**3.2.2 Quenching experiments in OH-initiated photooxidation**
To examine the contributions of ROS to eugenol degradation for OH-initiated
oxidation, TBA and $p$-BQ as trapping agents were added. Similar to $^3C^*$-initiated
oxidation, several gradient experiments with varying molar ratios of eugenol to
quenchers were conducted. The ratios were set as 6.5:1, 3.2:1, 1.6:1, 1.1:1 and 0.8:1 for
$p$-BQ and 3.0:1, 1.5:1, 0.75:1, 0.3:1 and 0.15:1 for TBA. According to Fig. S3, molar
ratio only had a slight influence on eugenol degradation, although degradation can be
inhibited effectively by quenchers. Thus, we determined the appropriate molar ratios of
0.8 and 0.75 for $p$-BQ and TBA, respectively, as excess scavengers might influence the
chemical reactions.
Variations of the rate constants for the aforementioned quenching experiments were
determined, in comparison with those conducted without quenchers, and results are
listed in Table 1 and presented in Fig. S4. For TBA quenching tests, the rate constant
decreased by 18.7% (from $2.73 \times 10^{-4}$ s$^{-1}$ to $2.22 \times 10^{-4}$ s$^{-1}$), showing that OH radical
played a certain role in eugenol photooxidation. Since $H_2O_2$ was mainly photolyzed at
wavelength <300 nm to generate OH radical, irradiation above 300 nm did not affect
the reaction significantly. The $p$-BQ could quench $O_2^{\bullet-}$, further suppressing the
generation of other ROS (e.g., $\bullet HO_2$), therefore the rate constant decreased the most
(from $2.73 \times 10^{-4}$ s$^{-1}$ to $1.20 \times 10^{-4}$ s$^{-1}$), suggesting $O_2^{\bullet-}$ was important for eugenol
photooxidation. This hypothesis could be further confirmed by the decline of rate
constant under $N_2$-saturated solution (Section 3.2.3). However, it was difficult to detect
both OH and $O_2^{\bullet-}$ directly due to their relatively short lifetimes and low concentrations
via ESR in this work.

**Table 1.** The first-order rate constants of eugenol in the presence of various scavengers. The initial conditions were as follows: 300 μM eugenol; molar ratios of eugenol to quenchers TBA, $NaN_3$, TMP and SOD, of 1.5, 0.15, 0.075 and 2.5, respectively; molar ratios of eugenol to quenchers $p$-BQ and TBA of 0.8 and 0.75, respectively.

| Quenchers | ROS | Reaction rate constant k ($s^{-1}$) | Pearson's $R^2$ |
|---|---|---|---|
| $^3C^*$-initiated quenching experiments | | | |
| no quencher | - | $5.75 \times 10^{-4}$ | 0.996 |
| TBA | OH | $2.65 \times 10^{-4}$ | 0.999 |
| SOD | $O_2^{\cdot-}$ | $2.22 \times 10^{-4}$ | 0.995 |
| $NaN_3$ | $^1O_2$ | $1.12 \times 10^{-4}$ | 0.999 |
| TMP | $^3C^*$ | $0.82 \times 10^{-4}$ | 0.999 |
| OH-initiated quenching experiments | | | |
| Quenchers | ROS | Reaction rate constant k ($s^{-1}$) | $R^2$ |
| No quencher | - | $2.73 \times 10^{-4}$ | 0.995 |
| TBA | OH | $2.22 \times 10^{-4}$ | 0.998 |
| $p$-BQ | $O_2^{\cdot-}$ | $1.20 \times 10^{-4}$ | 0.995 |

### 3.2.3 Influences of different saturated gases

In order to assess the role of $O_2$ in eugenol degradation, a series of experiments were performed under both $O_2$-saturated and $N_2$-saturated conditions in addition to air. $N_2$ gas was purged into reaction solution for ~30 minutes before experiment to achieve the $O_2$-free condition. Figure 3 compared the changes of eugenol concentrations and rate constants under the three gas conditions for direct photolysis, OH-initiated and $^3C^*$-initiated oxidations, respectively. The rate constants followed the order of $k_{O_2} > k_{Air} > k_{N_2}$ under both direct photolysis and OH oxidation, providing evidence in support of $O_2$ being significant for eugenol degradation. This might be explained by the fact that $O_2$ can act as an electron acceptor to generate $O_2^{\cdot-}$ and $\cdot HO_2$, and subsequently form $H_2O_2$ and OH. For direct photolysis, rate constant under $O_2$-saturated condition increased 14.4% while it decreased 19.3% under $N_2$ saturation from that under saturated air. For

OH-initiated oxidation, the difference of rate constants under three saturated gases
became more distinct.
On the contrary, rate constants followed the order of $k_{Air} > k_{N_2} > k_{O_2}$ in $^3C^*$-
initiated oxidation. There are two possible explanations. On one hand, under $N_2$-
saturated condition without oxygen, DMB would involve in reactions (R1-R4), leading
to a more effective generation of $^3DMB^*$ therefore a higher degradation efficiency than
under $O_2$-saturated condition. On the other hand, for air/$O_2$-saturated solutions,
irradiation of DMB and eugenol would involve also reactions (R5-R8) in addition to
(R1-R4), and as a result, the amount of $^3DMB^*$ decreased, due to formation of other
ROS ($^1O_2$, $O_2^{\bullet-}$, OH, etc) with relatively weak oxidative capacities. In summary,
quenching of $^3DMB^*$ by ground state molecular oxygen could account for the low
degradation efficiency in $O_2$-saturated condition.
$$DMB + h\nu \rightarrow {}^1DMB^* \rightarrow {}^3DMB^* \tag{R1}$$
$$^3DMB^* \rightarrow DMB \tag{R2}$$
$$^3DMB^* \rightarrow Products \tag{R3}$$
$$^3DMB^* + {}^1DMB^* \rightarrow DMB^{\bullet+/\bullet-}(DMB^{\bullet+}+DMB^{\bullet-}) \tag{R4}$$
$$^3DMB^* + O_2 \rightarrow DMB + {}^1O_2 \tag{R5}$$
$$DMB^{\bullet-} + O_2 \rightarrow \cdot DOM^+ + O_2^{\bullet-} \tag{R6}$$
$$O_2^{\bullet-} + 2H^+ \rightarrow H_2O_2 + O_2 \tag{R7}$$
$$H_2O_2 \rightarrow 2OH \tag{R8}$$

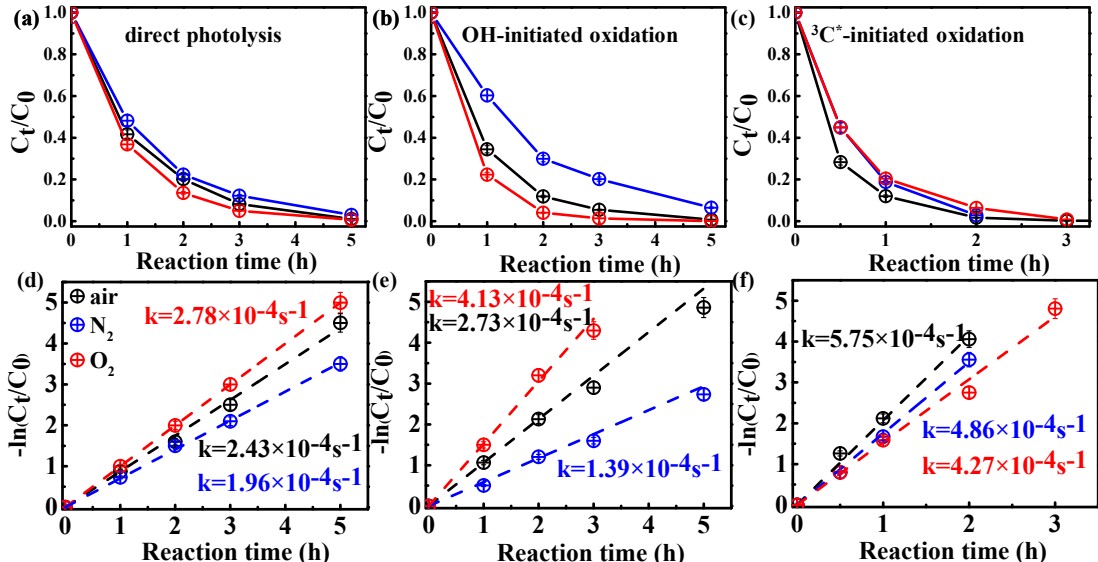

**Figure 3.** Ratio of unreacted eugenol concentration to its initial concentration ($C_t/C_0$) as a function of reaction time at different saturated gases under (a) direct photolysis (b) OH-initiated oxidation and (c) $^3C^*$-initiated oxidation. Rate constants of (a-c) are presented in (d-f) correspondingly.

### 3.2.4 Variations of pH and dissolved oxygen

The initial pH values of reaction solutions for direct photolysis and OH-initiated oxidation were unadjusted, while that for the $^3C^*$-oxidation was adjusted to 3. The variation of solution pH is presented in Fig. 4a. The pH values decreased quickly at the beginning of illumination (from 7.4 to ~5.0 in the first 1 hour) then tended to be flat for both direct photolysis and OH-initiated oxidation. However, little change of pH (less than 0.1 unit) was observed for the $^3C^*$-initiated photooxidation throughout the oxidation, which can be likely ascribed to its low initial pH of 3. Since the solution pH was acidic (pH=3), we cannot rule out formation of acidic products (such as organic acids) during $^3C^*$-initiated oxidation as during direct photolysis and OH-initiated oxidation.

As discussed in Section 3.2.3, oxygen can take part in photochemical reaction to form ROS, which may in turn destroy the structure of precursor. Here we measured the

oxygen consumption during oxidation through determination of dissolved oxygen (DO)
contents by a dissolved oxygen meter (Seven2Go Pro S9, Zurich, Switzerland). DO was
consumed mainly at the first 1 hour and remained stable afterwards (Figs. 4b-c and Fig.
S5). The amounts of consumed DO followed the order of $^3C^*$>OH>direct photolysis.
The maximum consumed DO was found in $^3C^*$-initiated oxidation, which might be
explained by the consumption of $O_2$ that reacts with $^3C^*$ form $^1O_2$ (R5). Obviously, a
steady-state DO level was reached when the consumption rate was equal to the diffusion
of $O_2$ into the solution (Pan et al., 2020). Overall, these results re-emphasize that $O_2$
can influence eugenol degradation and chemical transformation via induction of radical
chain reactions.

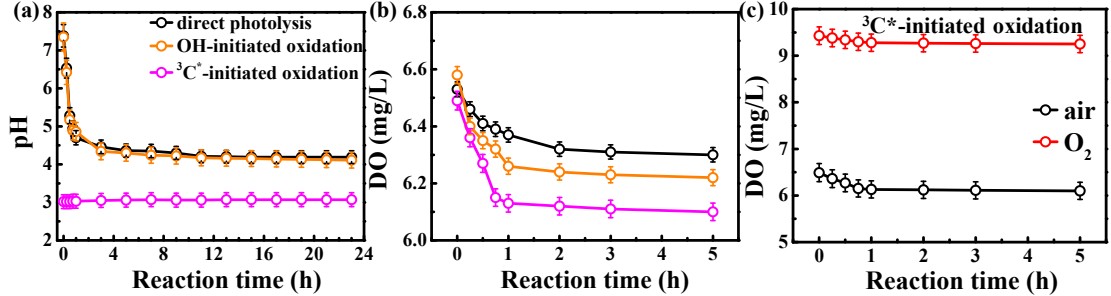


**Figure 4.** (a) pH values and (b) dissolved oxygen (DO) contents against reaction time under direct
photolysis, OH-initiated oxidation, $^3C^*$-initiated oxidation, and (c) DO contents during $^3C^*$-initiated
oxidation under air or $O_2$-saturated conditions.

## 3.3 Optical properties of reaction products

### 3.3.1 Light-absorbing properties

The UV-vis light absorption spectra of the solutions at different reaction times are
presented in Fig. 5. The light absorption by eugenol itself mainly occurs in the range of
260-300 nm (n→π* electronic transition, 270-350 nm), which overlaps with the major
photon fluxes (280 and 500 nm) of our lamp for photooxidation. Therefore, we can
clearly observe that the characteristic absorption peak at 280 nm of precursor decreased
with the propagation of direct photolysis (Fig. 5a), similar to that in OH-initiated
photooxidation (Fig. 5b). However, the reaction was quick in the presence of $^3C^*$, and
the characteristic absorption peak at 280 nm after 3 hours of illumination almost
disappeared, suggesting nearly a complete loss of eugenol, consistent with the results
in Section 3.1 that more than 99% eugenol was degraded in 3 hours. Additionally, there
was an obvious absorption enhancement at longer wavelengths (300-400 nm) during
the photooxidation, whereas eugenol itself did not absorb light in this range, indicating
some light-absorbing products (e.g., brown carbon (BrC) species) were generated.
Aqueous photooxidation of some phenolic compounds (e.g., vanillic acid) also
presented long-wavelength (300-400nm) light absorbance, with intensity increasing
with illumination time (Tang et al, 2020; Zhao et al., 2015). In addition, there were
some differences for light absorbance at wavelength of 300-400 nm in the three reaction
conditions. For direct photolysis and OH-initiated oxidation, light absorbance increased
during the first 15 hours, then remained at a plateau until 23 hours. However, for $^3C^*$-
initiated oxidation, light absorbance increased during the first 7 hours, then decreased
slowly afterwards. The different shapes of UV-vis spectra between OH and $^3C^*$
photooxidations indicate formations of different products.

Compared to the light spectrum of eugenol, there were also increases of light

absorbance at ~260 nm ($\pi \rightarrow \pi^*$ electronic transitions) upon aqueous oxidation in all
three reaction conditions (Fig. 5), demonstrating the generation of new substances
likely with both aromatic C=C and carbonyl (C=O) functional groups (Tang et al., 2020).
The enhancement at 300-400 nm may point to products with high MWs and conjugated
structures. Unfortunately, we were unable to quantify the relative contributions of
individual products to the overall light absorbance between 300 to 400 nm due to lack
of a full speciation of the products and their light absorption spectra.

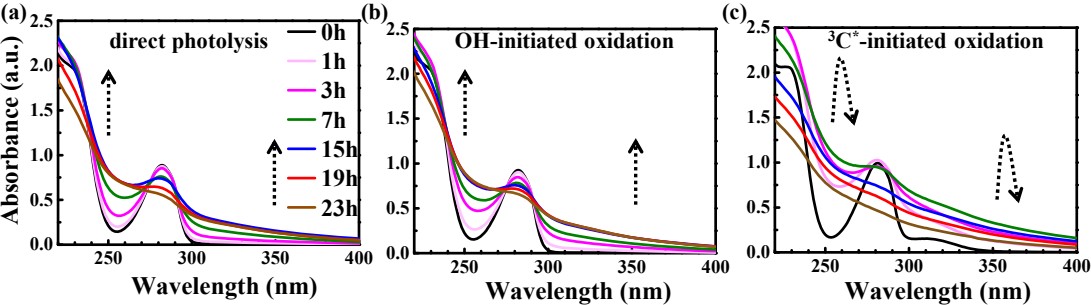


**Figure 5.** UV–vis light absorption spectra of reacted solutions at different reaction times under (a) direct photolysis, (b) OH-initiated oxidation, and (c)$^3$C$^*$-initiated oxidation.

## 3.3.2 Fluorescence properties

The fluorescence properties of solutions before (0 hour) and during photooxidation (3 and 7 hours) were investigated via the EEM technique, as shown in Fig. 6. For comparison, we also presented EEM profiles of pure eugenol (non-irradiated), pure DMB, and the end solutions (23 hours) of direct photolysis and OH-initiated oxidation in Fig. S6. The peaks at Excitation/Emission (Ex/Em)=275/313 nm can be attributed to fluorescence of the phenolic structure of parent substance (eugenol here), as suggested by Laurentiis et al. (2013). As shown in both Fig. 6 and Fig. S6, the fluorescence intensity decreased after oxidation due to eugenol decay, and the reduction was very fast for $^3$C$^*$-initiated oxidation. This finding matches with the fast degradation and large rate constant for $^3$C$^*$-initiated oxidation. The EEM plots for direct photolysis and OH-initiated oxidation had similar contour patterns as shown in Figs. 6a and b, although EEM profiles changed significantly with irradiation time. We also observed distinct fluorescent peaks at Ex/Em=235/(400-500) nm, indicating that illumination can cause a red shift in fluorescence emission wavelength. As suggested by Chang et al. (2010), fluorophores at Ex/Em=240/400 nm are linked with aromatic structures and condensed saturated bonds including polycyclic aromatic hydrocarbons. Another work (Li et al.,

2021) showed that red shift in the fluorescence spectra was usually related to an increase in the size of ring system and an increase in the degree of conjugation. Previous studies (Chen et al., 2016a; Chen et al., 2019) have reported that fluorescent compounds with emission wavelength at 400-500 nm were likely linked with HULIS. Additionally, HULIS have two typical fluorescent peaks in EEM profile at Ex/Em=(200-300)/(400-500) nm and Ex/Em=350/(400-500) nm with the former one having a higher intensity (Graber and Rudich, 2006; Laurentiis et al., 2013; Vione et al., 2019). There was also evidence that direct photolysis of tyrosine and 4-phenoxyphenol generated HULIS with new fluorescence signals at Ex/Em=(200-250)/(400-450) nm and 300/(400-450) nm (Bianco et al., 2014). In this regard, we inferred that new peak at Ex/Em=235/(400-500) nm here was likely attributed to HULIS. For the $^3C^*$-initiated oxidation, extra fluorescent peaks at Ex/Em=(220-300)/(400-500) nm appeared in the first 1 hour (data not shown), but their intensities weakened and gradually disappeared upon prolonged reactions (3 hours). Nevertheless, EEM results should be interpreted with caution because many substances might contribute to absorption and emission at a certain wavelength, and it is hard to distinguish and isolate fluorescent and nonfluorescent constituents simply via the EEM technique.

Another interesting finding was that a small fluorescence peak appeared at Ex/Em=(300-350)/(300-350) nm in some of the EEM profiles. Specifically, it appeared earlier for $^3C^*$-oxidation (at 3 hours) than the other two systems, yet its intensity seemed to be a bit stronger in the end solutions of direct photolysis and OH-oxidation (Fig. S6). Moreover, as suggested by Leenheer and Croue (2003), fluorescence peak position of the maximum Ex/Em for HULIS with lower MWs would shift towards lower wavelengths, thus, we inferred fluorescence peak at Ex/Em=(300-350)/(300-350) nm might be in part attributed to the organic acids with a few carbon atoms (probably $C_1$-

C$_6$). Nevertheless, large uncertainties still exist in using EEM fluorescence technique
to identify molecular compositions of the products due to lack of standard EEM profiles
for specific compounds from aqueous phase oxidation and clearly more studies are
needed in future.

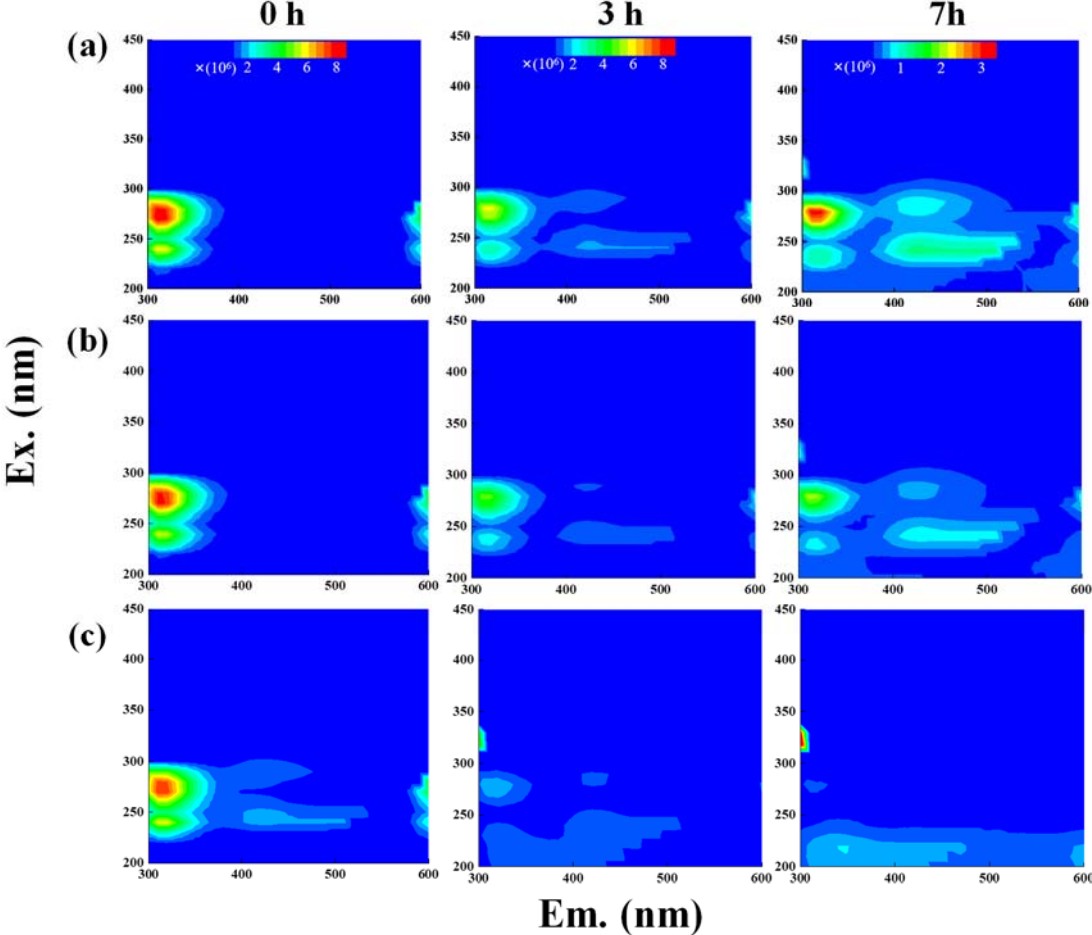


**Figure 6.** EEM fluorescence spectra of the initial solution (0 hour) and those at different reaction
time (3 and 7 hours) under (a) direct photolysis, (b) OH-initiated oxidation, and (c) $^3$C$^*$-initiated
oxidation.
**3.4 Characteristics of HULIS**

The EEM spectra revealed new prominent fluorescent peak at Ex/Em=250/(400-

500) nm, which was likely owing to HULIS. HULIS can be divided into fulvic acid
(water soluble at all pHs), humic acid (base soluble, acid insoluble) and humin
(insoluble at all pHs). In principle, extracted HULIS in this work with polymer-based
HLB SPE packing include LMW organic acids, fulvic acids and other humic substances.
Figure 7 presents the measured HULIS concentrations against the reaction time.
The results show clearly that aqueous-phase eugenol oxidation is a source of HULIS,
and the amount increased gradually in the first 7 hours, then remained at a similar level
(about 30 mg/L) for the OH-initiated oxidation. For direct photolysis, HULIS
concentration increased until 11 hours and then became steady at a level around 40
mg/L. For the $^3C^*$-oxidation, HULIS concentration increased to a maximum at 7 hours,
then declined slightly afterwards. A plausible reason of such variabilities is that
generated HULIS was capable of further taking part in photochemical reactions since
it can act as photosensitizer. Moreover, Yu et al. (2016) characterized the products from
aqueous oxidations of phenols by $^3C^*$ triplet states and OH radicals, and found both
could produce oligomers and hydroxylated species but the $^3C^*$-oxidation could produce
more of these compounds when 50% of the precursor was reacted. Considering the
large increases of HULIS in the first 7 hours and the much faster increase of $^3C^*$-
oxidation in the first 3 hours shown in Fig. 7, we postulate that HULIS species might
be some of the high MW oligomers, which can in turn contribute to fluorescence at
emission of ~400 nm (Barsotti et al., 2016).

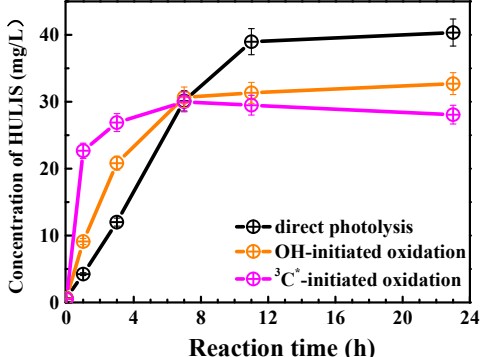


**Figure 7.** HULIS concentrations as a function of reaction time under direct photolysis, OH-
initiated oxidation and $^3C^*$-initiated oxidation.

**3.5 Mass yield and oxidation degree of reaction products**

**3.5.1 Mass yields**

HULIS is only a subset of the products from aqueous oxidation, and here we used AMS to further quantify the total reaction products. Figure 8a shows SP-AMS measured organic mass profiles (normalized by sulfate mass, $\Delta Org/SO_4^{2-}$) against the reaction time. As the reaction propagated, $\Delta Org/SO_4^{2-}$ increased continuously in $^3C^*$-initiated system. Nevertheless it arose stepwise and reached a maximum at 19 hours, then remained at a plateau for the direct photolysis and OH-mediated oxidation. Figure 8b illustrates the calculated mass yields at different reaction times. The mass yields were in the ranges of 46.2%-196.5%, 22.1%-144.9%, 19.3%-140.1% for $^3C^*$-oxidation, OH-oxidation and direct photolysis, respectively. For the same oxidation time, mass yield from $^3C^*$-oxidation was generally higher than those from OH-oxidation and direct photolysis. There are two plausible reasons for high mass yield of $^3C^*$-initiated oxidation. First, oxidation by $^3C^*$ was more efficient to form oligomers and functionalized/oxygenated products (Richards-Henderson et al., 2014; Yu et al., 2016). Higher oxidative degree of products from $^3C^*$-initiated photooxidation (see Sec.3.5.2) supports this hypothesis. Secondly, more light-absorbing products formed during initial stage of $^3C^*$-oxidation (Fig. 5c) may accelerate oxidation by acting as photosensitizers (Tsui et al., 2018).

The product mass yields obtained in this work (~20%-197%) overall agree with those reported previously for phenolic compounds. For examples, Huang et al. (2018) reported mass yields of 30-120% for syringaldehyde and acetosyringone; Smith et al. (2014) found that mass yields of aqSOA from three phenols with $^3C^*$ were nearly 100%, and Ma et al. (2021) reported a yield ranging from 59 to 99% for six highly substituted

phenols with $^3C^*$; Mass yields of SOA from three benzene-diols were near 100% with
both OH and $^3C^*$ oxidants (Smith et al., 2015); Direct photolysis of phenolic carbonyls,
and oxidation of syringol by $^3C^*$, had SOA mass yields ranging from 80 to 140% (Smith
et al., 2016). Our previous study on eugenol OH oxidation illuminated by a 500 W Xe
lamp reported a mass yield of ~180% (Ye et al., 2020), slightly higher than the value
determined here owing to different light wavelengths/intensities.

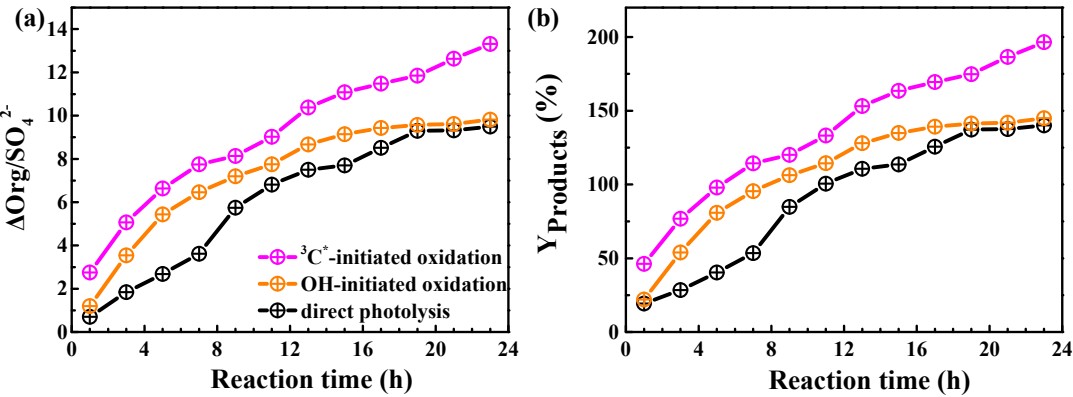


**Figure 8.** Variations of the organic mass normalized by sulfate (a) ($\Delta Org/SO_4^{2-}$) and (b) mass yields
of reaction products with reaction time under direct photolysis, OH-initiated oxidation and $^3C^*$-
initiated oxidation.

**3.5.2 Oxidation degree**

In order to further probe oxidation levels of the reaction products, O/C derived

from SP-AMS mass spectrum of the organics was used to represent the oxidation degree
of products. In addition, carbon oxidation state (OSc, defined as 2*O/C - H/C) (Kroll
et al., 2011) was also calculated. Figures 9a-c depict variations of the elemental ratios
(O/C and H/C) and OSc during oxidations. Rapid increases of O/C and OSc during the
initial stage of oxidation (within 1 hour) were observed, with O/C changing from 0.26
to 0.65, from 0.26 to 0.70, from 0.25 to 0.75, as well as OSc from -1.11 to -0.15, from
-1.16 to -0.05, from -1.13 to 0.09 for direct photolysis, OH-oxidation and $^3C^*$-oxidation,

respectively. The O/C was lower than those of other phenolic aqSOA (Yu et al., 2014) due to different substituted groups in aromatic ring of the precursors. Both O/C and OSc gradually increased, while H/C changed little after 1 hour. The enhancements of OSc in the end were 1.22, 1.11 and 0.86 for $^3C^*$-initiated oxidation, OH-initiated oxidation and direct photolysis, respectively.

Furthermore, the $f_{44}$ vs. $f_{43}$ diagram ("triangle plot") can be used to demonstrate the evolution of SOA during oxidation (Ng et al., 2010). The $f_{44}$ and $f_{43}$ are defined as the ratios of signal intensities of $m/z$ 44 (mainly $CO_2^+$) and 43 (mainly $C_2H_3O^+$) to the total organics. The results that the $f_{44}$ increased continuously (moved upwards) during both OH and $^3C^*$ oxidations, indicating persistent formation of highly oxygenated compounds including organic acids, such as formic acid and oxalic acid (Sun et al., 2010). Note the $f_{44}$ enhancement was much more significant for $^3C^*$ oxidation (from 0.07 to 0.16) than direct photolysis (from 0.07 to 0.12) and OH oxidation (from 0.07 to 0.13), consistent with its higher O/C and OSc. The $f_{43}$ value decreased in the first stage (1-3 hours) and then increased at later stages. The final $f_{43}$ values were almost the same as those of the initial solutions and were small. As a result, all data points located outside the $f_{44}$ vs. $f_{43}$ region (bounded by the two dash lines in Figs. 9d-f) for ambient aerosols established by Ng et al. (2010).

In summary, our results shown here demonstrate that aqueous phase eugenol photochemical oxidation can generate highly oxygenated products and hence increase the degree of oxygenation of overall SOA.

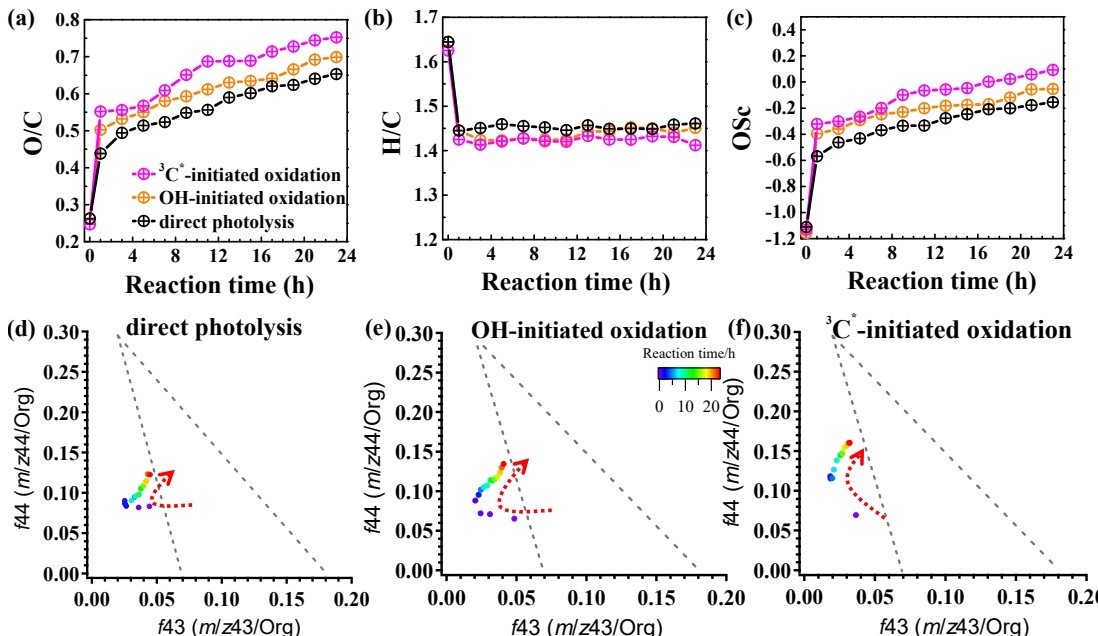

**Figure 9.** Variations of the elemental ratios of (a) O/C, (b) H/C and (c)oxidation state (OSc) as a function of reaction time; $f_{44}$ vs. $f_{43}$ plots of reaction products under (d) direct photolysis, (e) OH-initiated oxidation, and (f) $^3C^*$-initiated oxidation.

## 3.6 Molecular characterization of reaction products and proposed reaction mechanism

### 3.6.1 Major products identified by GC-MS

SP-AMS was limited to probe bulk composition of low-volatility oxidation products, thus the molecular-level characterization of products was performed by using GC-MS here. The total ion chromatograph (TIC) of GC-MS on the solutions before illumination (0 hour) and at illumination times of 11 and 23 hours for the $^3C^*$-initiated photooxidation is shown in Fig. S7. As shown in Fig.S7, eugenol (retention time (RT) at 11.50 min) loss was more than 90% at 11 hours, which could be confirmed by the experimental data reported in Section 3.1. Comparison of products at 11 hours and 23 hours showed no significant difference. Similar to aqueous photochemical oxidation with OH (Ye et al., 2020), a series of products were identified and listed in Table 2.

Except 5-ally-3-methoxybenzene-1,2-diol (MW 180, RT=12.59 min), the other eight
products were detected for both OH and $^3C^*$-initiated photooxidations. Some of them
(Eugenol, DMB, product 1, 2, 5) were identified by using certified reference materials,
some of them (product 3, 4, 6, 7, 8, 9) were inferred according to the molecular ion
peaks and fragments from GC-MS, based on spectra from the NIST database (Stein,
2014) and on the reactants and reaction conditions.
We also found 4-(1-hydroxypropyl)-2-methoxyphenol (product 8) was relatively
abundant (Fig.S7), suggesting functionalization might dominate as compared to
oligomerization and fragmentation. Products were mainly from addition/elimination of
hydroxyl (-OH), methoxyl (-OCH$_3$) to benzene ring or allyl group and further oxidized
to carbonyl or carboxyl compounds. As suggested by Bonin et al. (2007), the OH-
addition to the aromatic ring of phenol preferentially takes place at the ortho (48%) and
the para (36%) positions, leading to the formation of OH-adduct product 6 (5-allyl-3-
methoxybenzene-1,2-diol). Notably, dimers and ring-opening products were not
observed, but they cannot be excluded since they would be probably out of the detection
of GC-MS technique (Vione et al., 2014).
**Table 2.** Major reaction products identified via GC-MS

| | RT (min) | Name* | Proposed chemical structure | Chemical formula | Nominal MW (g/mol) |
|---|---|---|---|---|---|
| Product 1 | 10.68 | 4-allylphenol | | $C_9H_{10}O$ | 134 |
| Precursor | 11.50 | Eugenol | | $C_{10}H_{12}O_2$ | 164 |

| | | | | | |
|---|---|---|---|---|---|
| Product 2 | 11.81 | 4-hydroxy-3-methoxybenzaldehyde | | $C_8H_8O_3$ | 152 |
| Product 3 | 12.06 | (E)-2-methoxy-4-(prop-1-en-1-yl)phenol | | $C_{10}H_{12}O_2$ | 164 |
| Product 4 | 12.11 | 4-(hydroxymethyl)-2-methoxyphenol | | $C_8H_{10}O_3$ | 154 |
| Product 5 | 12.18 | 2-methoxy-4-propylphenol | | $C_{10}H_{14}O_2$ | 166 |
| Photosensitizer | 12.29 | 3,4-dimethoxybenzaldehyde(DMB) | | $C_9H_{10}O_3$ | 166 |
| Product 6** | 12.59 | 5-allyl-3-methoxybenzene-1,2-diol | | $C_{10}H_{12}O_3$ | 180 |
| Product 7 | 12.65 | 4-(1-hydroxyallyl)-2-methoxyphenol | | $C_{10}H_{12}O_3$ | 180 |
| Product 8 | 12.79 | 4-(1-hydroxypropyl)-2-methoxyphenol | | $C_{10}H_{14}O_3$ | 182 |
| Product 9 | 12.91 | (E)-4-(3-hydroxyprop-1-en-1-yl)-2-methoxyphenol | | $C_{10}H_{12}O_3$ | 180 |

*Precursor (eugenol) and triplet precursor (DMB) are also shown.
**This compound was only identified in $^3$C*-oxidation solution.
**3.6.2 Reaction mechanism**
The reaction pathways of $^3$C*-initiated photooxidation of eugenol are
demonstrated in Scheme 1 based on the products identified by GC-MS. The
other intermediates and the potential pathways were proposed according to the
identified products and the reaction rationality from the starting reactant. To better
depict the mechanism, DMB was expressed as [RCHO] and eugenol as Ph-R for
simplicity. [RCHO] absorbs light and undergoes excitation to $^1$[RCHO]$^*$, then
experiences the intersystem crossing (ISC) to form $^3$[RCHO]$^*$. $^3$[RCHO]$^*$ can
participate in subsequent reactions via three channels. First, it can react with $O_2$ to form
$^1O_2$ via energy transfer. Secondly, it can transform to [RCHO]$^{•-}$, subsequently reacts
with $O_2$ to generate $O_2^{•-}$ via electron transfer, which can disproportionate to $H_2O_2$. The
decomposition of $H_2O_2$ can generate OH radical. Thirdly, the $^3$[RCHO]$^*$ can react with
Ph-R to from [Ph-R•] via H-abstraction. The cleavage of [Ph-R•] to free radical segment
(such as $CH_2CH•$ or $CH_3O•$) takes place, then an additional hydrogen transfer could
occur, resulting in a 2H-addition to the new intermediate to form 4-allyl-phenol
(product 1). Similarly, when the $CH_2CH•$ is lost from [Ph-R•], an addition of $H_2O$
would happen on the new compound (product 4) and further oxidized to 4-hydroxy-3-
methoxybenzaldehyde (product 2). Another possibility is the intermediate [Ph-R•] can
resonate to several different isoelectronic species, the radical position changes to
aromatic ring or allyl group site, which would couple with OH to form hydroxylated
eugenol monomer (product 6, 7, 9 MW=180). Consequently, the isoelectronic species
at allyl group site could also abstract a hydrogen to form isoeugenol (product 3
MW=164). Also, breakage of C=C into C-C and 2H-addition at allyl group site could

form 2-methoxy-4-propyl-phenol (product 5, MW=166). Besides, the C=C breaking intermediate can couple with OH to form 4-(1-hydroxypropyl)-2-methoxyphenol (product 8, MW=182). In conclusion, $^3C^*$ can directly oxidize eugenol to form SOA products or small molecular compounds, or indirectly oxidize eugenol via energy transfer, electron transfer, hydrogen abstraction, proton-coupled electron transfer or other radical chain reactions.

The organic groups, such as methoxy, allyl groups can be eliminated from aromatic ring, which then participate in photochemical reaction, resulting in generation of dimers, small organic acids, $CO_2$ and $H_2O$, etc. Dimers previously reported from aqueous reaction of 4-methylsyringol with OH were not detected via GC-MS in the present work but dimer fragment ions ($C_{20}H_{22}O_4^+$) were detected by SP-AMS with trace amounts. Functionalization due to the additions of hydroxyl, carbonyl functional groups to the aromatic rings could account for the enhancement of light absorption at wavelength of 300-400 nm. However, polar high MW organic acids were not detected likely due to the limitation of GC-MS technique.

**Scheme 1.** Proposed reaction mechanism of $^3C^*$-initiated photooxidation of eugenol. The red text represents the precursor, and the compounds labeled by Product 1-9 are those identified by GC-MS (Table 2).

## 3.7 Oxidative potential (OP) of reaction products

The OP of oxidation products can be represented by the consumption rate of DTT concentration, defined as $R_{DTT}$. Figure 10a shows the DTT consumed mass ($M_{DTT}$) as a function of incubation times (0, 30, 60, 90, 120 and 150 min) for a triplicate sample

(300 μM eugenol) and blank (ultrapure water). $M_{DTT}$ values for both blank and eugenol
were proportional to incubation time, indicating that ROS-generating substances in
reaction solution act only as catalyst and itself was not consumed. The slopes represent
DTT consumption rates, which are also illustrated in Fig. 10a. Average $R_{DTT0}$ (blank)
was 0.31 μM/min and $R_{DTT}$ for initial 300 μM eugenol (before experiment) was 0.52
μM/min. Since self-oxidation of DTT might lead to the consumption of DTT in
ultrapure water, final DTT consumption rate of reacted solution after oxidation was then
blank-corrected by subtracting the average $R_{DTT0}$.

729   Figure 10b shows changes of blank-corrected $R_{DTT}$ with reaction time for direct

photolysis, OH-initiated oxidation and $^3C^*$-initiated oxidation, respectively. The $R_{DTT}$
value of $^3C^*$-oxidation products increased quickly and reached the maximum (0.9) at 7
hours, then decreased slowly and its end value was lower than that from OH-oxidation.
The $R_{DTT}$ value of OH-oxidation products on the other hand increased slowly and
reached the maximum at 21 hours. The $R_{DTT}$ value of products from direct photolysis
increased continuously but also slowly to ~0.36 till the end of oxidation. Nevertheless,
we can see that the final $R_{DTT}$ values were all higher than that of eugenol, proving that
aqueous-phase processing can generate products with higher OP, resulting in more
health hazards than the precursor does. The DTT consumption rates are comparable to
those using the same DTT method (Charrier and Anastasio, 2012; Lin and Yu, 2019).
The weak correlation was found between HULIS concentration and $R_{DTT}$, implying that
OP was not only dependent upon HULIS. Moreover, HULIS with diverse molecular
structures also exhibit different ROS-generation potentials (Kramer et al., 2016),
therefore the HULIS as an ensemble may not correlate well with OP.

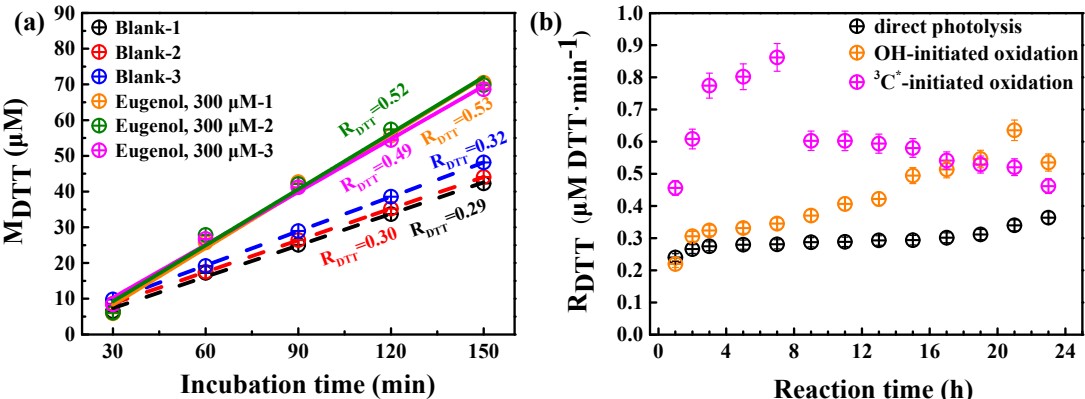

**Figure 10.** (a) DTT consumed mass versus incubation times for blank (ultrapure water) and 300 μM eugenol solutions in a triplicate, and (b) blank-corrected DTT consumption rates versus reaction time for direct photolysis, OH-initiated oxidation and $^3C^*$-induced oxidation.

## 4. Atmospheric implications

The high mass yields of aqueous-phase photooxidation of eugenol (exceeding 100% after 23 hours of illumination) found here are similar or even higher than those previously reported yields of a number of phenolic compounds (e.g., Smith et al., 2014, 2015, 2016; Ma et al., 2021), which re-emphasizes the importance of biomass burning (BB) to SOA budget (Gilardoni et al., 2016), particularly in regions or periods with significant BB activities. In addition, our study here used 300 μM $H_2O_2$ and 15 μM DMB as sources of OH and $^3C^*$, and $^3C^*$-mediated oxidation appeared to be faster than OH-initiated oxidation of eugenol. Of course, whether or not $^3C^*$ is more important than OH in real atmosphere depends upon their concentrations. OH and $^3C^*$ are difficult to measure and concentrations vary greatly in real atmospheric samples. Herrmann et al. (2010) estimated an average OH level of $0.35 \times 10^{-14}$ M in urban fog water; Kaur and Anastasio (2018) measured $^3C^*$ concentration to be $(0.70\text{-}15) \times 10^{-14}$ M, 10-100 times higher than the co-existing OH in ambient fog waters; Kaur et al. (2019) determined both OH and $^3C^*$ concentrations in PM extracts, OH steady-state concentration was

4.4($\pm$2.3) x10$^{-16}$ M, similar to its level in fog, cloud and rain, while $^3$C* concentration
was 1.0($\pm$0.4) x10$^{-13}$ M, a few hundred times higher than OH and nearly double its
average value in fog. Therefore, together with these measurements, our findings signify
a likely more important role of $^3$C* than OH in aqueous-phase (especially aerosol water)
reactions. However, the liquid water content of aerosol is typically ~10000 times
smaller than that of cloud (for instance, ~50 $\mu$g m$^{-3}$ versus 0.5 g m$^{-3}$). Even if the
reaction rates in aerosol water were 10 times higher than those in cloud water, the
overall importance of aqueous reactions initiated by the same oxidant in aerosol phase
would be still ~1000 times smaller than it in cloud water. Moreover, quenching
experiments reveal that O$_2$ can inhibit eugenol degradation by effectively scavenging
$^3$C* while it can promote degradation by fostering chain reactions in OH-induced
oxidation, which offer insights to the control of reaction pathways by regulating ROS
generations; of course, such operation calls for application of highly sensitive EPR
method.
Eugenol has a strong light absorption peak around 280 nm, therefore it can undergo
direct photolysis, and addition of OH or other photosensitizers ($^3$C*) can gradually
diminish its light absorption around 280 nm, but increase the absorption in visible light
range (>300 nm). In the meantime, HULIS was generated continuously, and GC-MS
identified a number of high MW organic products, in line with those detected in earlier
aqueous photooxidation of phenolic compounds (Jiang et al., 2021; Misovich et al.,
2021; Tang et al., 2020; Yu et al., 2014). Overall, our work demonstrates that aqueous
oxidation of BB emissions is a source of BrC, and this BrC may act as photosensitizer
to oxidize other species; a portion of this BrC might be HULIS, and some high MW
aromatic compounds are a subset of this HULIS. However, a recent study by Wang et
al. (2021) shows that fossil fuel derived OA (FFOA) can be an effective precursor of

aqSOA, but the aqSOA became less light-absorbing than the FFOA. These contrasting results indicate that contribution of aqueous oxidation to BrC is largely dependent upon the precursors; molecular structures of major chromophores, changes of the structures upon oxidation as well as their interplay with light absorptivity should be carefully investigated to achieve a full understanding of the impacts of aqueous processing on air quality, radiative forcing and climate change.

Investigations on the OPs of reaction products from eugenol photooxidation show that aqueous processing can produce more toxic products than the precursor. This result is in agreement with our previous work on resorcinol, hydroquinone and methoxyhydroquinone (Ou et al., 2021). Although more studies on a broad spectrum of atmospherically relevant species and multiple indicators of toxicity are clearly needed, our findings here underscore the potential of aqueous processing on the enhancement of particle toxicity.

## 5 Conclusions

This study comprehensively investigated the aqueous photooxidation of eugenol upon direct photolysis and attacks by OH radicals and $^3C^*$ triplet states. By using a suite of techniques, the decay kinetics of eugenol, chemical, optical properties as well as toxicity of reaction products were studied. The first-order rate constants followed the order of $^3C^*$>OH >direct photolysis (300 μM $H_2O_2$ and 15 μM DMB as sources of OH and $^3C^*$). Further quenching experiments on different ROS during $^3C^*$-mediated oxidation showed that $^3C^*$ was the major contributor, followed by $^1O_2$, $O_2^{•-}$ and OH; $O_2^{•-}$ played a more important role than OH during OH-initiated oxidation. The rate constants under saturated $O_2$, air and $N_2$ followed the order of $k_{O_2} > k_{Air} > k_{N_2}$ for both direct photolysis and OH-initiated oxidation, but changed to $k_{Air} > k_{N_2} > k_{O_2}$ for $^3C^*$-

mediated oxidation. $O_2$ appeared to be a scavenger of $^3C^*$ therefore suppressing $^3C^*$
oxidation while it could promote generation of OH thus accelerate OH-mediated
oxidation. pH and DO levels both decreased during oxidation, indicating formation of
acids and a certain role of DO in oxidation.

Eugenol itself can absorbs lights significantly around 280 nm, and aqueous

oxidation gradually decrease this absorption of UV light but enhanced the absorbance
in the visible light range (mainly 300-400 nm), indicative of the generation of BrC
species. These species were likely linked with HULIS, as HULIS concentration
increased during the course of oxidation, in particular for the initial stage of $^3C^*$-
mediated reactions. The final mass yields of reaction products (after 23 hours of
irradiation) were 140.1%, 144.9% and 196.5% for direct photolysis, OH-oxidation and
$^3C^*$-oxidation, respectively. Oxidation degrees of the products increased continuously
with the illumination time, indicating persistent formation of highly oxygenated
compounds, especially during $^3C^*$-mediated reactions. Molecular characterization by
GC-MS identified a series of oxygenated compounds, allowing us to propose the
detailed oxidation mechanism. Functionalization appeared to be a dominant pathway to
form the observed species.

DTT method was used to assess OP of the reaction products. The end products in

all three sets of experiments showed higher DDT consumption rates than that of the
precursor; products from $^3C^*$-oxidation showed particularly fast increase in the first
few hours of reactions. This result demonstrates that species that are more toxic than its
precursors could be produced upon aqueous oxidation, indicative of the potential toxic
effects induced by aqueous processing.

***Data availability.*** The relevant data of this study are available at:
http://nuistairquality.com/eugenol_data_and_figure

*Supplement.* The supplement related to this article is available on line at: XXX

*Author Contributions:* XDL, YT, LWZ, SSM, SPL, ZZZ and NS conducted the
experiments. XDL and YT analyzed the data. XDL and ZLY prepared and wrote the
paper with contributions from all co-authors. ZLY and XLG reviewed and commented
on the paper.

*Competing interests.* The authors declare that they have no conflict of interest.

*Acknowledgements.* The authors acknowledge support from the National Natural
Science Foundation of China (21976093 and 42021004), the Natural Science
Foundation of Jiangsu Province (BK20181476), open fund by Jiangsu Key Laboratory
of Atmospheric Environment Monitoring and Pollution Control (KHK1904) and the
Postgraduate Research & Practice Innovation Program of Jiangsu Province
(SJCX21_1332, SJCX20_1030) and of Jiangsu University of Technology
(XSJCX20_05).

*Financial support:* This research was funded by the National Natural Science
Foundation of China (21976093 and 42021004), the Natural Science Foundation of
Jiangsu Province (BK20181476), and open fund by Jiangsu Key Laboratory of
Atmospheric Environment Monitoring and Pollution Control (KHK1904).

*Review statement.* This paper was xxx

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
