# Peer review of "Optical properties and oxidative potential of aqueous-phase products"

_Atmospheric Chemistry and Physics, 2021_

## Author Comment (AC1)

**A letter to reply the comments**

**Title: Optical properties and oxidative potential of aqueous-phase products from OH and $^3C^*$-initiated photolysis of eugenol (acp-2021-895)**

**Dear Editor**

We would like to very much thank you and the reviewers for the comments regarding our manuscript. These comments are important to help us to improve our work. We have carefully revised the manuscript accordingly. A thorough grammar and spelling check of the manuscript were also done. Our point-to-point responses to the reviewers' comments are listed below.

**Reviewer #1**

This reviewer enjoyed reading this manuscript, which presents a quite systematic investigation of the aqueous-phase photo-oxidation of eugenol. Indeed, these authors conducted a comprehensive analysis of the degradation kinetics of eugenol, along with the chemical-, optical-properties as well as toxicity (oxidative potential) of the products under direct photolysis, OH- and $^3C^*$-initiated oxidation in the bulk aqueous-phase. The manuscript is well structured and each subsection comes with a thoughtful discussion. I would therefore recommend its publication subject to minor corrections.

**Response:**

1.One of the key messages here is that the $^3C^*$-initiated oxidation dominates over the OH one. This is interesting and in line with other reports. However, it should nice to state whether this is only valid for the actual laboratory conditions selected here, or if it can be also extrapolated to atmospheric conditions. In other words, how does the ratio of the $^3C^*$ to OH radical concentrations compare to realistic atmospheric conditions? Can the authors comment on that and therefore justify their choices of concentrations for the oxidant precursors?

**Response:** We thank very much the overall positive view on our manuscript. In our previous work (Atmospheric Environment 223 (2020) 117240) about aqueous photochemical reaction of phenolic compounds with hydroxyl radical under the same condition, we detected steady-state concentration of OH radical of about $6 \times 10^{-12}$ M throughout the course of sunlight OH experiment, which was one order of magnitude higher than that in cloud environment, but close to that in wet aerosol microenvironment. Concentration of DMB as $^3C^*$ source was determined according to references (Environ. Sci. Technol. 2021, 55, 5199−5211; Phys. Chem. Chem. Phys., 2015, 17, 10227; Atmos. Chem. Phys., 16, 4511–4527, 2016 ).

2.The use of acronyms for the selected compounds is justified but it does not ease the reading of the manuscript. It would be nice to state the actual of the compounds when firstly mentioned in the result section (and not just in the experimental one).
**Response:** Thanks for your attention. We have paid attention this issue and revised in this new manuscript.

3. While this manuscript reads well, a final polishing of the English would improve it further (but this is a very minor comment).
**Response:** In addition, we have made major revisions especially added more discussion for this manuscripts to make it easy to follow. We hope the new version meets similarity requirement.

**Reviewer #2**

In this study the authors investigated the degradation of eugenol by three ways i.e. photolysis, excited triplet states, and OH radicals. The obtained results suggest that the excited triplet states are most reactive toward eugenol followed by OH radical induced degradation and photodegradation. The fluorescent spectra indicated formation of HULIS. The use of dithiothreitol indicated formation of harmful species during the oxidation process. I think that this topic could be interesting for the readers of ACP and the manuscript should be reconsidered upon major revision. I have some comments that I hope can be helpful for the authors to improve the quality of the manuscript. My main comments are mostly related to the experimental details.

**Response:** We thank very much the overall positive view of the reviewer on my manuscript. We appreciate very much the useful comments raised by the reviewer below and we have tried our best to address them point-by-point as appended below.

**Comments:**

1.The authors used mercury lamps with discontinuous emission spectrum at 313 nm, 365nm, 419nm, and 436 nm to irradiate the aqueous solution consisting of eugenol, which can potentially lead to misleading conclusions. For this kind of experiments, it is more appropriate to use Xenon lamp (solar simulator) with continuous emission spectrum from 300 to 700 nm. It will be useful in Figure S1 to compare the spectral irradiance from the mercury lamp with the sunlight spectral irradiance.Figure 5a: How it is caused the photolysis of eugenol that absorbs light at 280 nm with mercury lamp irradiating at 313, 365, 419 and 465 nm?

**Response:** We are very appreciated for your suggestion. Actually, Rayonet photoreactor (model RPR-200) in this work (See photo below) was frequently used for photochemical reaction and described in detail by several groups (George et al., 2015; Hong et al., 2015; Huang et al., 2018; Jiang et al., 2021; Zhao et al., 2014) to mimic sunlight. The photoreactor and light sources were the same as above mentioned. The normalized distribution of the photon fluxes inside reactor have been reported elsewhere (George et al., 2015, see below Figure). According to their description, the wavelength of photon fluxes was over 280 and 500 nm range. In order to avoid mistaking, we deleted Fig. S1.

[Figure]

Normalized distribution of the photon fluxes inside our RPR-200 illumination system, and, for comparison, the fluxes from our solar simulator and for midday, winter solstice, actinic flux in Davis, CA(George et a.., 2015, Atmos. Environ., 100, 230-237)

2. The changes of pH values and the levels of dissolved oxygen before and after the reactions in the aqueous phase are not reported.

**Response:** We have supplement Section 3.2.4 "Variation of pH value and dissolved oxygen (DO)" in the revised manuscript. In this section, we monitored pH value and DO concentration upon photolysis process, which were shown in Fig.4 and Fig. S5. Additionally, we analyzed the results.

[Figure]

**Figure 4.** pH values and dissolve oxygen as a function of reaction time for the three systems.

3. Line 88: it should state "several" instead of "sever".Figure S3: On X axis should be "magnetic field" instead of "magntic field". Figure 3: It is not correct to state "under OH system". Please revise it.

**Response:** Done.

4. Line 251: Which chemical bond?

**Response:** We have reorganized Section 3.1 "Kinetics of the photo-oxidation" and embellish language in order to express clearly. Chemical bond has been replaced by bond dissociation energies (BDE). So, this sentence "The energies of photons at 313, and 365 nm are 395 kJ/mol, and 338 kJ/mol, which are higher than certain chemical bond energies, for instance, 354 kJ/mol for C-C, and…" has been deleted.

5. Line 255: In a real world environment the photon energy of the sunlight does not have two peaks at 313 nm and 365 nm.

**Response:** We totally agree with you. Rayonet photoreactor (model RPR-200) in this work was frequently used for photochemical reaction and described in detail by several groups According to their description, the wavelength of photon fluxes was over 280 and 500 nm range. So, the spectra exhibited a broader range of wavelength. We inferred our spectra determination in the original manuscript has not been corrected.

6. Lines 375-376: To state that the "role of OH is weak" is not appropriate term here. Moreover, the statement is somewhat misleading as the continuous emission spectrum would probably induce different effect.

**Response:** Thanks. This sentence has been optimized to "As seen in Fig. 5, when adding oxidant $H_2O_2$, the total variation trend of light absorbance was similar to that without oxidant with some slight difference, for instance at wavelength range of 200-250 nm."

7. Line 389: "the formation of "brown carbon". Did authors observed change of the colour of the solution? Did really becomes brownish?

**Response:** Thanks. We did not observed obvious change of colour of solution. Thus, This sentence has been changed to "The difference plot of UV-Vis spectra between OH and $^3C^*$-initiated photo-oxidation indicated the formation of different products." to make sense. Furthermore, we added some description in the last paragraph "The increase of light absorbance at 250 nm upon aqueous photo-processing demonstrates the generation of new substances with both the aromatic C=C and carbonyl (C=O) functional groups, while the enhancement at 300-400 nm suggested the probability of HULIS formation, which could be confirmed later."

8. Please change the title of section 2.3.5. "Products analysis of products" is not right.

**Response:** Done.

9. Lines 421-422: Please rewrite the sentence starting with "This is likely due......". It is not clear at all the meaning of this sentence.

**Response:** Yes. This sentence has been changed to "This could be attributed to different precursor and aqueous reaction mechanisms (Xie et al. 2016). "

10. My other comments are related to the references. The authors used many self-citations and at first glance the paper looks like that previously not so many groups studied the reactions of phenols and methoxyphenols in the aqueous phase. For example, lines 57-61: Many recent papers related to the photodegradation of phenolic substances are not cited here.

**Response:** Thanks for your suggestion. We referred to more articles (Gilardoni et al., 2016; He et al., 2019; Jiang et al., 2021; Li et al., 2014; Li et al, 2021; Mabato et al., 2022; Tang et al., 2020; Yang et al., 2021; Yu et al., 2016) published recently and related to reactions of phenolic carbonyl compounds in aqueous-phase.

11. Lines 66-67: There are other more appropriate references related to OH radical concentrations in the atmospheric aqueous phase. See for example: Chem. Rev. 2015, 115, 24, 13051–13092; Chem. Rev. 2003, 103, 4691-4716.

**Response:** Thanks. We have cited suggested representative paper with regard to OH radical concentration in the atmospheric aqueous phase.

12. Line 93: In which sense these compounds can damage human body? Also reference is here needed about the health implications of quinones and PAHs.

**Response:** Thanks for your suggestion. We have **rewritten "Section 1: Introduction",** for example: Earlier report from Chang and Thompson (2010) found fluorescence spectra of reaction products during aqueous reaction of phenolic compounds, with some similarities with aerosol Humic-like substances (HULIS); Tang et al.(2020) also observed light-absorbing products formed in aqueous-phase OH oxidation of vanillic acid and further verified that aqueous reaction was a potential source of HULIS. Recently, Li et al (2021) began to apply EEM technique to characterize formation of light-absorbing compounds in aqueous phase oxidation of syringic acid. Additionally, previous studies (Chang and Thompson, 2010) showed that light-absorbing and fluorescent substances generally have large conjugated moieties (i.e., quinones, HULIS, polycyclic aromatic hydrocarbons (PAHs)), which can damage human body (McWhinney et al., 2013; Dou et al., 2015). HULIS are considered as an important contributor to induce oxidative stress since they can served as electron carriers to catalyze ROS formation. Dithiothreitol (DTT) assay (Alam et al., 2013; Chen et al., 2021;Verma et al., 2015), as a non-cellular method, was widely employed to determine oxidation activity and assess oxidative potential of atmospheric PM via the rate of DTT consumption (Cho et al., 2005; Chen et al., 2019), since oxidative stress was related to adverse health effect. You can see we cited many references. Sentence **"HULIS are considered as an important contributor to induce oxidative stress since they can served as electron carriers to catalyze ROS formation"** can indicate mechanism of damaging human body by HULIS.

13. Lines 251-255: Please give the reference for the reported bond dissociation energies (BDE) which is by the way more appropriate term than chemical bonds energies.

**Response:** In Section 3.1, we added some description about BDE in detail: The BDEs are 340 kJ/mol for OH, 374 kJ/mol for C-H in $-CH_3$ group, 345 kJ/mol for C-C in C=C bonds, and 403 kJ/mol for C-H in $-OCH_3$ group, respectively (Herrmann et al., 2003;He et al., 2019). The lowest BDE was found for the O-H bond and C-C bond. Due to the influence of steric hindrance and intramolecular hydrogen bonding, the H-abstraction reaction from the OH group might have been less favorable. The most favorable H-abstraction reaction might have taken place in the C-C in allyl group. As a result of breakage of C=C into C-C at allyl group site, 2-methoxy-4-propyl-phenol could form (See Section 3.6.1). When photon energy is higher than bond dissociation energy, they can directly break chemical bond of molecules, leading to decomposition of compounds and possibly further mineralization. The energies of photons at 313, and 365 nm in our

light sources are 395 kJ/mol, and 338 kJ/mol, which are higher than the weakest BDEs in eugenol, as a result, eugenol molecule can directly absorb photo energy to decompose.

14. Lines 406-409: There are many previous studies that have shown the formation of fluorescent compounds associated with HULIS at wavelengths 400-500nm. For example: http://dx.doi.org/10.1016/j.atmosenv.2013.09.036, doi.org/10.1016/j.atmosenv.2019.03.005)

**Response:** Thanks. We have cited references (Laurentiis et al., 2013; Vione et al., 2019; Wu et al., 2021) to illustrate the formation of fluorescent of HULIS at 400-500nm.

The manuscript presents an interesting investigation on the aqueous phase photooxidation of eugenol, and its impacts on light-absorption and oxidative potential. However, the entire manuscript is presented as a report, very far from the standard requests of a scientific publication. It lacks organization and a logical follow up : instead of clearly providing UV-vis spectra of eugenol and comparing it to the solar actinic fluxes and to their lamp's actinic fluxes to show the potential importance of the photolysis reaction, the authors directly start with a complex comparison between the kinetics of photolysis vs OH-oxidation vs reactivity towards $^3C^*$, with no appropriate discussion (see below).

**Response:** We have reorganized the whole paper. We have made major revisions especially add more discussion for this manuscripts to make it easy to follow.

1.The use of SI is not appropriate: the most interesting figures are provided in SI, and redundant figures are provided in the text with less information (see for example Figure 2 compared to Figure S2)

**Response:** Thanks for your suggestion. We removed some interesting figures from supplementary materials to the text (i.e., Figure 2 and Figure 3). We added Table 1 to make rate constant under different quenchers more clear. Furthermore, some figures with less important information were placed in Supplementary material. Additionally, we added e changes of pH values and the levels of dissolved oxygen with reaction time.

**Table 1.** The reaction rate constants of eugenol in the presence of scavengers. The experimentl conditions were as follows: 0.3 mM eugenol, molar ratios of eugenol to quencher TBA, $NaN_3$, TMP and SOD, of 1.5, 0.15, 0.075 and 2.5 respectively; mole ratio of eugenol to quencher $p$-BQ and TBA of 0.8 and 0.75 respectively.

| $^3C^*$-initiated quenching | | | |
|---|---|---|---|
| quenchers | ROS | reaction rate constant k $(s^{-1})$ | $R^2$ |
| no quencher | - | $5.75 \times 10^{-4}$ | 0.996 |
| TBA | OH | $2.65 \times 10^{-4}$ | 0.999 |
| SOD | $O_2^-$ | $2.22 \times 10^{-4}$ | 0.995 |
| $NaN_3$ | $^1O_2$ | $1.12 \times 10^{-4}$ | 0.999 |
| TMP | $^3C^*$ | $0.82 \times 10^{-4}$ | 0.999 |
| $\cdot$OH-initiated quenching | | | |
| quenchers | ROS | reaction rate constant k $(s^{-1})$ | $R^2$ |
| No quencher | - | $2.73 \times 10^{-4}$ | 0.995 |
| TBA | OH | $2.22 \times 10^{-4}$ | 0.998 |
| $p$-BQ | $O_2^-$ | $1.20 \times 10^{-4}$ | 0.995 |

[Figure]

**Figure 2.** Ratio of residue concentration to initial concentration ($C_t/C_0$) at different mole ratios as a function of reaction time with (a) TBA quencher, (b) NaN$_3$ quencher, (c) TMP quencher and (d) SOD quencher. Legend represented mole ratios of eugenol to quenchers.

[Figure]

**Figure 3.** Ratio of remaining concentration to initial concentration ($C_t/C_0$) as a function of reaction time at different saturated gases under (a) direct photolysis (b) OH-initiated and (c) $^3C^*$-initiated oxidation. Insert plots: Plots of eugenol consumption versus reaction time under different saturated gases: (a) direct photolysis (b) OH-initiated and (c) $^3C^*$-initiated systems.

2.The authors provide a systematic and very convincing study of quenching reactions during the photolysis of eugenol, but they forget to discuss on their analytical uncertainties when comparing the influence of the various quenchers. This is particularly critical for the discussions on Fig S4 and S5a.

**Response:** We agree with you. Firstly, we have rewritten **Section 3.2** "Relative importance of ROS to photo-oxidation" and added more detailed description. For each

scavenger, we conducted several gradient experiments with varying molar ratios of eugenol to quenchers. Tanking C3-iniitiated reaction for example. The ratios were set as 0.075:1, 0.15:1, 0.3:1, 0.75:1, 1.5:1 for quenchers of $NaN_3$, TMP and TBA, and 1.2:1, 1.6:1, 2.5:1, 5:1, 10:1 for SOD, which were all within the typical range of molar ratios to quench ROS reported previously (Zhou et al., 2018). Above concentrations of the added quencher have been repeatedly adjusted to ensure the complete reactions between radicals and scavengers. The optimum molar ratios of eugenol to quenchers were selected when the inhibition degree of eugenol degradation unchanged with the increase of added quencher mass (Wang et al., 2021). For example, upon decreasing molar ratios of eugenol to $NaN_3$ from 1.5:1 to 0.075:1, the inhibitory degree of eugenol degradation was unchanged at ratio of 0.15:1 and 0.075:1, indicating that $^1O_2$ has been absolutely quenched at ratio of 0.15:1, so, we finally selected molar ratios of 0.15:1 for $NaN_3$, since excess scavenger may produce other products that can change the existing reaction. Finally, the molar ratios of eugenol to quencher TBA, $NaN_3$, TMP and SOD, of 1.5, 0.15, 0.075 and 2.5, were selected, respectively. So, the uncertainties when comparing the influence of the various quenchers can be reduced. Additionally, for most series of experiments, solution was saturated by air and all experiments presented were conducted in triplicate unless otherwise stated. The results were shown in respect of average plus/minus standard deviation. Moreover, in Section 2.3.4, we added "Experiments of blanks and samples were typically run in a triplicate. The reproducibility of the whole analysis showed that the relative standard deviation of the DTT consumption rate analysis was 3-4%."

3.Too many results are presented with no clear links, and sometimes with no appropriate discussion, but only in reference to other papers, mostly from their own group. For example, the comparison between the kinetics of photolysis vs OH-oxidation vs reactivity towards $^3C^*$ where no discussion is performed on the amount of OH and $^3C^*$ precursors, neither the lamp's actinic flux.

**Response:** We have cited many references to make some conclusion sense. For example, we cited (Gilardoni et al., 2016; He et al., 2019; Jiang et al., 2021; Li et al., 2014; Li et al, 2021; Mabato et al., 2022; Tang et al., 2020; Yang et al., 2021; Yu et al., 2016) published recently, related to reactions of phenolic carbonyl compounds in aqueous-phase; References (Laurentiis et al., 2013; Vione et al., 2019; Wu et al., 2021) were cited to indicate EEM spectra at Em of 400-500nm; The amounts of radicals can be identified and quantified by the peak patterns in EPR spectra, such as quarter line with a height ratio of 1:2:2:1 for DMPO-•OH, 1:1:1:1 for DMPO-$O_2^{•-}$ and 1:1:1 for TEMP-$^1O_2$ (Guo et al., 2021);

Actually, Rayonet photoreactor (model RPR-200) in this work (See photo below) was frequently used for photochemical reaction and described in detail by several groups (George et al., 2015; Hong et al., 2015; Huang et al., 2018; Jiang et al., 2021;

Zhao et al., 2014) to mimic sunlight. The photoreactor and light sources were the same as above mentioned. The normalized distribution of the photon fluxes inside reactor have been reported elsewhere (George et al., 2015, **see below Figure**). According to their description, the wavelength of photon fluxes was over 280 and 500 nm range. In order to avoid mistaking, we deleted Fig. S1.

[Figure]

Normalized distribution of the photon fluxes inside our RPR-200 illumination system, and, for comparison, the fluxes from our solar simulator and for midday, winter solstice, actinic flux in Davis, CA(George et a.., 2015, Atmos. Environ., 100, 230-237)

4.Many small errors can be mentioned. See for example confusion between water solubility and water miscibility, confusion between a radical and triplet state species, …There are too many English errors and typos;

**Response:** Sorry for our .carelessness. In revised manuscript, we have corrected typographical errors, grammatical inaccuracies, misspelling, and so on, we hope the new version meets the journal's requirement.

---

## Editor Decision (ED1)

Several major concerns remain that were raised by the previous referees. I summarize them below, together with a detailed list of additional comments. Please clearly indicate in your response where and how you have addressed the comments in the revised manuscript.

**I) Major comments**

1) I still share Referee #3's concern that the manuscript reads like a lab report rather than a scientific paper. The various experiments are listed but the connection between the experiments and their conclusions are not fully clear.
It would help if you referred to the various subsections within Section 3 to connect your results (e.g. can the kinetics be related to the reaction mechanism, how does the production of HULIS relate to the reaction mechanism etc?)

2) A lack of a coherent discussion was also expressed by Referee #1 who asked you to comment on the atmospheric relevance of the oxidant concentrations and their ratios.
Therefore, I suggest that you add a separate section on the discussion of the atmospheric relevance of your results. Please also include a brief discussion on the representativeness of eugenol, and how its reactivity can inform about SOA processing (formation and loss) by precursors of intermediate volatility.

3) You discuss the formation of HULIS and it seems at some places that you exchangeable use "HULIS' with aqSOA. Please define what HULIS are in your discussion and how 'humic-like' and atmospherically-relevant these products formed in the reactions are (cf. e.g. discussion by (Graber and Rudich, 2006)). Please check carefully your discussion of HULIS and how comparable the products in your experiments are to HULIS as referred to in the literature (e.g. line 672).

**II. Minor comments**

1) l. 37/38: Keywords are not needed

2) l. 58: I share the referee #2's concerns that you do not properly cite previous literature on aqueous phase oxidation of organics, in particular of phenolic compounds. A huge number of kinetic and product studies for small VOCs and also for phenolic compounds have been performed before 2010 (the oldest currently cited paper here). Please cite also relevant older references of the aqueous phase oxidation of phenolic compounds. See for example review articles (Herrmann, 2003; Herrmann et al., 2015) and specific other studies (Barzaghi and Herrmann, 2002; Bonin et al., 2007; Sun et al., 2010)

3) l. 80/81: There are numerous studies that clearly state the role of ROS (including OH, H2O2) for SOA formation. Please include appropriate literature or revise this sentence.

4) l. 260: A 'rate' refers to a speed of a process; therefore, it has units of 1/time or concentration/time. Please clarify what $\eta$ is. I assume you mean 'degree of degradation' (dimensionless)(?)

5) l. 319-338: This text needs to be clarified and further elaborated on (see also Referee#1's comment 1). It needs to be made clear whether these findings of the relative importance are only valid under your experimental conditions or also to the atmosphere.

6) Section 3.5.1: How do the aqSOA yields compare to those reported in other studies, e.g. (Ma et al., 2021; Smith et al., 2014)? Why do you find different values than in your previous study?

7) Section 3.6.2: How quantitative is this reaction mechanism? Were all intermediates and products identified by GC-MS or were some inferred?

**III. Technical comments**

Please carefully proofread your manuscript and correct unclear language, grammar mistakes and typos. I added a long, but not exhaustive list below.

l. 18: 'under two radicals' should be 'in the presence of radicals'

l. 22: define 'ESP'

l. 45: replace 'are' by 'is' (aqSOA is singular)

l. 46: either 'less volatile' or 'have lower volatility'

l. 51: 'not yet' is redundant

l. 61/62: This sentence is not clear.

l. 63: I assume you do not mean $^1O_2$ (singlet oxygen) and not molecular oxygen.

l. 66: 'predominant oxidant' is ambiguous. It is correct that the rate constants of OH reactions are usually much higher than those of the NO3 radical or other oxidants. However, other oxidants (such as H2O2 or $^1O_2$ (singlet oxygen) might be equally high or even higher)

l. 93/94: Why can vanillic acid be considered a proxy for HULIS?

l. 100/101: (1) Please add a reference to this sentence. (2) replace 'served' by 'serve'.

l. 107: replace 'has' by 'have'

l. 120/121: What do you mean by 'deeply clarifying the degradation mechanism'?

l. 142/3: 'At the bottom of sample tubes, there are fan and  magnetic stir bar to make solution full mixed' – reword, e.g. 'To ensure mixing of the solution, a fan and a magnetic bar are placed at the bottom of the solution'.

l. 150/1: 'slightly lower' is not correct, if you refer to light intensity that is more than 60% lower than sun light.

l. 159: replace 'darkness' by 'dark'

l. 161: replace 'now' by 'not'

l.201: replace 'was' by 'were'

l. 222: 'for distinguish' – do you mean 'as a blank'? Please clarify.

l. 232: replace 'Products' by 'Product'

l. 245: 'CHO$_1$' should be 'CHO'

l. 280: Unclear what 'they' refers to. Are you referring to chemical bonds in general or to a specific bond referred to in the previous sentence?

l. 285: Please reword this sentence, e.g., 'The pseudo-first-order rate constants were obtained by fitting eugenol concentration to the equation [please add equation number here!]. The experiments were performed under conditions of excess oxidants.'

l. 290/1: I do not understand this sentence: Are you saying that the oxidation by 3C* is faster because it is a combination of multiple pathways including reactions with 1O2, O2- and OH? Please clarify.

l. 301: What does 'they' (were calculated) refer to? If you mean 'the relative importance', it should be 'it'.

l. 349: replace 'combing' by 'combining'

l. 371 – 373: This sentence is not clear. Please clarify.

l. 371: Do you really mean 'photolysis' here or 'photooxidation'?

l. 374: add 'by' (decreased by 56%).

Figure 3: The labels next to the lines in all panels are really hard to read. Please improve the figure quality.

l. 420: replace 'directly' with 'direct'

l. 424 – 426: This sentence is not clear. Please clarify.

l. 439: replace 'In a word' by 'In summary'

l. 497: fluorescent

l. 498: replace 'more intense' by 'higher intensity'

l. 504: replace 'photosensitise' by 'photosensitizer experiments'

l. 510 – 517: What is the main message of this text? Please reword and clarify.

l. 519: 'characterize aqSOA' is too general. Do you mean molecular composition or structure, optical properties, ...?

l. 500; l. 527: HULIS

l. 592 – 594: Please correct and clarify this sentence.

l. 644: This sentence seems redundant.

**References**

Barzaghi, P. and Herrmann, H.: A mechanistic study of the oxidation of phenol by $OH/NO_2/NO_3$ in aqueous solution, Phys. Chem. Chem. Phys., 4, 3669–3675, 2002.

Bonin, J., Janik, I., Janik, D. and Bartels, D. M.: Reaction of the Hydroxyl Radical with Phenol in Water Up to Supercritical Conditions, J. Phys. Chem. A, 111(10), 1869–1878, doi:10.1021/jp0665325, 2007.

Graber, E. R. and Rudich, Y.: Atmospheric HULIS: how humic-like are they? A comprehensive and critical review, Atmos. Chem. Phys., 6, 729–753 [online] Available from: http://www.atmos-chem-phys.net/6/729/2006/acp-6-729-2006.pdf, 2006.

Herrmann, H.: Kinetics of aqueous phase reactions relevant for atmospheric chemistry, Chem. Rev., 103(12), 4691–4716, 2003.

Herrmann, H., Schaefer, T., Tilgner, A., Styler, S. A., Weller, C., Teich, M. and Otto, T.: Tropospheric Aqueous-Phase Chemistry: Kinetics, Mechanisms, and Its Coupling to a Changing Gas Phase, Chem. Rev., 115(10), 4259–4334, doi:10.1021/cr500447k, 2015.

Ma, L., Guzman, C., Niedek, C., Tran, T., Zhang, Q. and Anastasio, C.: Kinetics and Mass Yields of Aqueous Secondary Organic Aerosol from Highly Substituted Phenols Reacting with a Triplet Excited State, Environ. Sci. Technol., 55(9), 5772–5781, doi:10.1021/acs.est.1c00575, 2021.

Smith, J. D., Sio, V., Yu, L., Zhang, Q. and Anastasio, C.: Secondary Organic Aerosol Production from Aqueous Reactions of Atmospheric Phenols with an Organic Triplet Excited State, Environ. Sci. Technol., 48(2), 1049–1057, doi:10.1021/es4045715, 2014.

Sun, Y., Zhang, Q., Anastasio, C. and Sun, J.: Insights into secondary organic aerosol formed via aqueous-phase reactions of phenolic compounds based on high resolution mass spectrometry, Atmos. Chem. Phys., 10, 4809–4822, 2010.

---

## Author Response (AR2)

**A letter to reply the comments**

**Title: Optical properties and oxidative potential of aqueous-phase products from OH and $^3$C$^*$-initiated photolysis of eugenol** (acp-2021-895)

**Dear Editor**

   We would like to very much thank you for the comments regarding our manuscript. These comments are very helpful for us to improve our paper. We have carefully revised the manuscript accordingly. A thorough grammar and spelling check of the manuscript were also done. Our point-to-point responses to your comments are listed below.

**Response for major comments:**

1)   I still share Referee #3's concern that the manuscript reads like a lab report rather than a scientific paper. The various experiments are listed but the connection between the experiments and their conclusions are not fully clear. It would help if you referred to the various subsections within Section 3 to connect your results (e.g. can the kinetics be related to the reaction mechanism, how does the production of HULIS relate to the reaction mechanism etc?)

**Response:** Thanks. We enhanced some links among experimental results, mechanism and products in revised manuscript. For instance, in Section 3.2.4 "Thus, the decrease of pH value might be related to formation of organic acid and HULIS since carboxylic acids are possibly abundant in HULIS (Huo et al., 2021; Salma et al., 2008)."; Section 3.4 "Thus, we inferred that more higher molecular weight HULIS were formed at the first stage due to oligomerization and functionalization as suggested by Jiang et al.(2021), which consequently contributed to fluorescence at Ex of 400 nm (Barsotti et al., 2016).", Section 4 "Significant absorption enhancement over the range of near-UV region (300-400 nm) pointed out the continuous generation of BrC (i.e., HULIS). Direct HULIS concentration determination confirmed that HULIS was formed continuously over the course of reaction. Consequently, these light-absorbing products contribute to fluorescence at EX of 400-500 nm. GC-MS analysis confirmed the formation of high molecular weight multi-functional organic compounds, which has also been reported previously in similar aqueous phenolic photochemical experiments (Jiang et al., 2021; Misovich et al., 2021; Tang et al., 2020; Yu et al., 2014). Overall, our work shows that

SVOCs-aqSOA is an important source of BrC, therefore aqueous chemical processes may play a role in aerosol light absorption, radiative forcing, as well as climate change. In-depth molecular-level characterization or functional groups with respect to HULIS should be carried out in the future study". Furthermore, we also reorganized Section 3.6.1 in order to establish connect between product and mechanism.

2)    A lack of a coherent discussion was also expressed by Referee #1 who asked you to comment on the atmospheric relevance of the oxidant concentrations and their ratios. Therefore, I suggest that you add a separate section on the discussion of the atmospheric relevance of your results. Please also include a brief discussion on the representativeness of eugenol, and how its reactivity can inform about SOA processing (formation and loss) by precursors of intermediate volatility.

**Response:** We have supplement atmospheric relevance in Section 4 "Conclusions and atmospheric implication" in the revised manuscript. Furthermore, in Sect. 1 "Introduction", we added some description about precursor eugenol. "In the present work, we choose 4-allyguaiacol/eugenol, as a model compound to conduct aqueous phase reaction. Eugenol, a representative methoxyphenol emitted from biomass burning (Hawthorne et al., 1989; Simpson et al., 2005), was widely detected in atmospheric particles. For instance, emission concentration and factor emitted from beech stove were 0.032 μg/m$^3$ and 1.534 μg/g, which are twice of those of (0.016 μg/m$^3$ and 0.762 μg/g) guaiacol (Bari et al., 2009; Liu et al., 2019). Eugenol is a representative semivolatile aromatic compounds with moderate water-solubility (2.46 g/L at 298 K)."

3)    You discuss the formation of HULIS and it seems at some places that you exchangeable use HULIS with aqSOA. Please define what HULIS are in your discussion and how 'humic-like' and atmospherically-relevant these products formed in the reactions are (cf. e.g. discussion by (Graber and Rudich, 2006)). Please check carefully your discussion of HULIS and how comparable the products in your experiments are to HULIS as referred to in the literature (e.g. line 672).

**Response**: Thanks for your attention. We have paid attention this issue and revised it in this new manuscript. In section 3.4, we referred some discussion by Graber and Rudich. "The EEM spectra found new prominent fluorescent peak at Ex/Em=250 nm/400-500 nm, which was likely attributed to chromophores of HULIS. Humic substances are subdivided into fulvic acid (water soluble at all pHs), humic acid (base soluble, acid (pH 1) insoluble) and humin (insoluble at all pHs). In principle, extracted HULIS with

polymer-based HLB SPE packing included LMW organic acids, fulvic acids or other humic substances. As suggested by Graber and Rudich (2006), two distinct ranges have been found to characterize humic substances: Ex/Em=330–350/420–480 nm (fulvic-like), and Ex/Em=250–260/380–480nm (humic-like). So, we inferred most HULIS in this paper was humic-like substance rather than fulvic-like substance."

**Response for minor comments:**
1) l. 37/38: Keywords are not needed
**Response:** Deleted.

2) l. 58: I share the referee #2's concerns that you do not properly cite previous literature on aqueous phase oxidation of organics, in particular of phenolic compounds. A huge number of kinetic and product studies for small VOCs and also for phenolic compounds have been performed before 2010 (the oldest currently cited paper here). Please cite also relevant older references of the aqueous phase oxidation of phenolic compounds. See for example review articles (Herrmann, 2003; Herrmann et al., 2015) and specific other studies (Barzaghi and Herrmann, 2002; Bonin et al., 2007; Sun et al., 2010)
**Response:** We have cited many relevant older references of the aqueous phase oxidation of phenolic compounds.

3) l. 80/81: There are numerous studies that clearly state the role of ROS (including OH, H2O2) for SOA formation. Please include appropriate literature or revise this sentence.
**Response:** This sentence has been changed to "Despite strong evidence in support of the importance of ROS in photochemical process (Ma et al, 2021;Wang et al., 2020; Wang et al., 2021; Wu et al., 2021), however, our understanding on SOA from $^3C^*$-initiated reaction is still limited."

4) l. 260: A 'rate' refers to a speed of a process; therefore, it has units of 1/time or concentration/time. Please clarify what $\square$ is. I assume you mean 'degree of degradation' (dimensionless)(?)
**Response:** Thanks for your attention. "degradation rate" has been replaced by "degradation efficiency".

5) l. 319-338: This text needs to be clarified and further elaborated on (see also Referee#1's comment 1). It needs to be made clear whether these findings of the relative importance are only valid under your experimental conditions or also to the atmosphere.

**Response:** Thanks for your suggestion. Other studies (Laurentiis et al., 2013; Misovich et al., 2021) of aqueous DMB-photosensitized reaction also showed $^3C^*$ was the greatest contributor to phenol or guaiacyl acetone loss, followed by $^1O_2$, while both •OH and $^1O_2$ contributions were relatively minor(Section 3.2.1).

6) Section 3.5.1: How do the aqSOA yields compare to those reported in other studies, e.g. (Ma et al., 2021; Smith et al., 2014)? Why do you find different values than in your previous study?

**Response:** We have reorganized Section 3.5.1: The SOA mass yield are slightly higher than value (ranging from 80-140%) from phenol-triplet reaction (Ma et al., 2021; Smith et al., 2016; Yu et al., 2014). For the same oxidation time, mass yields from $^3C^*$oxidation were generally higher than those from OH-initiated oxidation and direct photolysis. These results were similar to investigation on aqueous oxidation of phenolic compounds (Smith et al., 2014, 2015, 2016). There are two plausible reason for high masses for $^3C^*$-initiated oxidation. Firstly, oxidation by $^3C^*$ was more efficiently to form oligomers and functionalized/oxygenated products (Richards-Henderson et al., 2014). Higher oxidative degree of aqSOA from $^3C^*$-initiated photooxidation (see Sec.3.5.2) warrants above hypothesis. Secondly, more light-absorbing products (i.e. HULIS) can participate in SOA formation by acting as photosensitizers (Tsui et al., 2018).

7) Section 3.6.2: How quantitative is this reaction mechanism? Were all intermediates and products identified by GC-MS or were some inferred?

**Response:** All the products (colored and numbered) were identified by using •GC-MS. Some of them (Eugenol, DMB, product 1, 2,5) were identified by using certified reference materials, some of them (product 3, 4, 9)identified by matching the NIST database, and others (product 6, 7, 8) were inferred according to the molecular ion peak and fragments from GC-MS, combine with NIST matching results and the start material & reaction conditions. The other intermediates and the mechanism were inferred and proposed according the identified products and the reaction rationality from start material. Corresponding description has been added in Section 3.6.1 and 3.6.2. All the

MS spectrums for Eugenol, DMB and products 1~9 were listed as following.

[Figure]

**Fig 1. Eugenol, t = 11.50 min**

[Figure]

**Fig 2.DMB , t = 12.29 min**

[Figure]

**Fig 3. Product 1, t = 10.68 min**

[Figure]

**Fig 4. Product 2, t = 11.81 min**

[Figure]

Fig 5. Product 3, t = 12.06 min

[Figure]

Fig 6. Product 4, t = 12.11 min

[Figure]

**Fig 7. Product 5, t = 12.18 min**

[Figure]

**Fig 8. Product 6, t = 12.59 min**

文件 : C:\Users\Administrator\Desktop\Pper-丁香酚-9.24\丁香酚产物
数据\GCMS原始数据\300uEUG+sunlight\300uEUG+15uC3_1h.D
操作员 :
仪器; GCMS
已采集 : 16 May 2020 15:27 , 使用采集方法 QZX-20180706-TIC.M
样品名称 : 300uEUG+15uC3_1h
其他信息 :

[Figure]

Fig 9. Product 7, t = 12.65 min

文件 : C:\Users\Administrator\Desktop\Pper-丁香酚-9.24\丁香酚产物
数据\GCMS原始数据\300uEUG+sunlight\300uEUG+15uC3_1h.D
操作员 :
仪器; GCMS
已采集 : 16 May 2020 15:27 , 使用采集方法 QZX-20180706-TIC.M
样品名称 : 300uEUG+15uC3_1h
其他信息 :

[Figure]

Fig 10. Product 8, t = 12.79 min

[Figure]

Fig 11. Product 9, t = 12.91 min

**III. Technical comments**

Please carefully proofread your manuscript and correct unclear language, grammar mistakes and typos. I added a long, but not exhaustive list below.

**Response:** Thanks for your suggestions. We have rewritten some paragraphs and made major revisions especially add more discussion for this manuscripts to make it easy to follow. We have also corrected typographical errors, grammatical inaccuracies, misspelling, and so on, we hope the new version meets the journal's requirement.

l. 18: 'under two radicals' should be 'in the presence of radicals'

l. 22: define 'ESP'

l. 45: replace 'are' by 'is' (aqSOA is singular)

l. 46: either 'less volatile' or 'have lower volatility'

l. 51: 'not yet' is redundant

**Response:** Done.

l. 61/62: This sentence is not clear.

**Response:** This sentence has been optimized to "Generally, chemical structure of precursors has significant influence on aqSOA and reaction mechanisms, however, the effect of oxidant on SOA formation also cannot be neglected.".

l. 63: I assume you do not mean 1O2 (singlet oxygen) and not molecular oxygen.

**Response:** Yes. We mean singlet oxygen and replaced it.

l. 66: 'predominant oxidant' is ambiguous. It is correct that the rate constants of OH reactions are usually much higher than those of the NO3 radical or other oxidants. However, other oxidants (such as H2O2 or 1O2 (singlet oxygen) might be equally high or even higher)

**Response:** We totally agree with you. 'predominant'has been changed to "ubiquitous" to make sense.

l. 93/94: Why can vanillic acid be considered a proxy for HULIS?

**Response:** This sentence has been changed to 'Subsequently, numerous studies have observed light-absorbing products formed in aqueous photodegradation and further verified that aqueous reaction was a potential source of HULIS (Li et al., 2021; Smith et al., 2016; Tang et al., 2020)'.

l. 100/101: (1) Please add a reference to this sentence. (2) replace 'served' by 'serve'.

**Response:** Done.

l. 107: replace 'has' by 'have'

**Response:** Done.

l. 120/121: What do you mean by 'deeply clarifying the degradation mechanism'?

**Response:** This sentence has been optimized to 'The relative importance of various ROS species to eugenol degradation was explored in order to clarify reaction mechanism.'

l. 142/3: 'At the bottom of sample tubes, there are fan and magnetic stir bar to make solution full mixed' – reword, e.g. 'To ensure mixing of the solution, a fan and a magnetic bar are placed at the bottom of the solution'.

**Response:** Thanks. We have revised according to your suggestion.

l. 150/1: 'slightly lower' is not correct, if you refer to light intensity that is more than 60% lower than sun light.

**Response:** 'slightly lower' has been replaced by 'was lower'.

l. 159: replace 'darkness' by 'dark'

l. 161: replace 'now' by 'not'

l.201: replace 'was' by 'were'

**Response:** Done.

l. 222: 'for distinguish' – do you mean 'as a blank'? Please clarify.

**Response:** Yes. We mean distinguish between blank and solution reaction. So, we rewritten this sentence: Another 1.2 mL ultrapure water instead of sample solution was treated in the same way and absorbance was denoted as A for distinguish from $A_t$.

l. 232: replace 'Products' by 'Product'

l. 245: 'CHO1' should be 'CHO'

**Response:** Done.

l. 280: Unclear what 'they' refers to. Are you referring to chemical bonds in general or to a specific bond referred to in the previous sentence?

**Response:** Thanks. We added 'As we known' before this sentence.

l. 285: Please reword this sentence, e.g., 'The pseudo-first-order rate constants were obtained by fitting eugenol concentration to the equation [please add equation number here!]. The experiments were performed under conditions of excess oxidants.'

**Response:** Thanks. We moved this sentence to the upper of the first paragraph in Section 3.1. And this sentence has been optimized to 'Figure 1 shows unreacted eugenol concentrations ($c_t$) and the negative logarithm of $c_t/c_0$ ($-\ln(c_t/c_0)$) as a function of reaction time, respectively. The pseudo first-order rate constant (k) obtained by Eq.(1) was also presented.'.

l. 290/1: I do not understand this sentence: Are you saying that the oxidation by 3C* is faster because it is a combination of multiple pathways including reactions with 1O2, O2- and OH? Please clarify.

**Response:** Yes. This sentence has been optimized to '$^3C^*$-initiated photodegradation was quicker than that with •OH due to contributions of combination of multiple pathways including reactions with $^1O_2$, $O_2^-\cdot$ and $\cdot OH$.'

l. 301: What does 'they' (were calculated) refer to? If you mean 'the relative importance', it should be 'it'.

**Response:** Yes. 'they were' has been replaced by 'it was'.

l. 349: replace 'combing' by 'combining'

**Response:** Done.

l. 371 – 373: This sentence is not clear. Please clarify.

**Response:** Yes. This sentence has been optimized to 'The *p*-BQ could quench $O_2^{\bullet-}$, which further suppress the generation of other ROS (e.g., •HO2), as a result, the rate constant decreased the most (from $2.73\times10^{-4}\,s^{-1}$ to $1.20\times10^{-4}\,s^{-1}$), suggesting $O_2^{\bullet-}$ might be responsible for eugenol photodegradation'.

l. 371: Do you really mean 'photolysis' here or 'photooxidation'?

**Response**:Yes. 'photolysis'was replaced by'photooxidation.

l. 374: add 'by' (decreased by 56%).

Figure 3: The labels next to the lines in all panels are really hard to read. Please improve the figure quality.

**Response**:Done.

l. 420: replace 'directly' with 'direct'

**Response:** Done.

l. 424 – 426: This sentence is not clear. Please clarify.

**Response:** The first paragraph in Section 3.2.4 has been optimized to 'As shown in Fig.4a, solution pH values decreased quickly at the beginning of illumination (from 7.4 to ~5.0 for the first 1h) then tended to smooth in both direct photolysis and OH-initiated oxidation. However, little change of pH value (less than 0.1 unit) was observed for the $^3C^*$-initiated photooxidation, which could be ascribed to very low initial pH value (pH=3). Generally speaking, slight increase of acidity cannot remarkably change pH value when original solution pH was very low. We cannot rule out formation of acid products at $^3C^*$-initiated oxidation. Thus, the decrease of pH value might be related to formation of organic acid and HULIS since carboxylic acids are possibly abundant in HULIS (Huo et al., 2021; Salma et al., 2008).'.

l. 439: replace 'In a word' by 'In summary'

l. 497: fluorescent

l. 498: replace 'more intense' by 'higher intensity'

l. 504: replace 'photosensitise' by 'photosensitizer experiments'

**Response:** Done.

l. 510 – 517: What is the main message of this text? Please reword and clarify.

**Response:** Thanks. We have reword this section.

l. 519: 'characterize aqSOA' is too general. Do you mean molecular composition or structure, optical properties?

**Response:** Yes. We changed 'characterize aqSOA' to 'characterize molecular composition'.

l. 500; l. 527: HULIS

**Response:** Done.

l. 592 – 594: Please correct and clarify this sentence.

**Response:** This sentence has been optimized to 'The O/C was lower than that (0.85-1.23) of other phenolic aqSOA reported due to different substituted group in aromatic ring (Yu et al., 2014).'

l. 644: This sentence seems redundant.

**Response:** Thanks. We deleted this sentence.

---

## Editor Decision (ED2)

**1) HULIS vs aqSOA vs oligomers**
As I pointed out previously, not all aqSOA is HULIS and not all HULIS is aqSOA. Therefore, it cannot be used interchangeably. In addition also oligomers and brown carbon are separate products. While there might be overlaps between these compounds groups (aqSOA, HULIS, BrC, oligomers) they all different definitions and should not be used equally.
To clarify and simplify the terminology throughout the paper, I suggest using consistently 'products', 'reaction products' or 'aqueous-phase products'.

l. 94: Did all three studies cited here indeed claim that the investigated reactions yield HULIS? Unfortunately, I do not have access to all of them; however, based on the abstracts, it seems to me that Smith et al., 2016 and Li et al., 2021 only refer to aqSOA or other products. Please verify the terminology in the original literature.
It is not clear how you differentiate between aqSOA and HULIS. Using the

l. 30 vs l. 247 vs l. 601 vs l. 633: While in the abstract you call it HULIS, in the experimental section, you call the products sometimes aqSOA, and sometimes HULIS. Please use consistently only one term (preferably 'reaction products') to avoid confusion.

l. 437f: Here you refer to studies on HULIS in ambient particulate matter. The HULIS found in these samples are not (necessarily) originating from reactions as investigated in the current study. Therefore, your products (aqSOA) are not comparable to these ambient HULIS.

l. 482: 'High molecular weight and conjugated structures' are also present in oligomers. Not all oligomers are HULIS. However, if these oligomers are formed by aqueous phase reactions, they can be termed aqSOA.

l. 550: Here you could simply delete 'of HULIS'.

l. 554 – 560: This text is out of place here and would be redundant if you use 'products'. Also it incorrectly cites the review by Graber and Rudich. 'HULIS' is the acronym for 'HUmic-LIke Substances'. Therefore, the sentence 'So, we inferred most HULIS in this paper was humic-like substance rather than fulvic-like substance' does not make sense.
Graber and Rudich discuss that "HULIS relate to the water soluble fraction, which would include only the fulvic acid fraction of humic substances, and exclude the humic acid (base-soluble) and humin (insoluble) fractions of humic substances."

l. 596: not all light-absorbing compounds are HULIS; and not all HULIS are light absorbing

l. 771: HULIS and brown carbon are not always the same.

**2) Atmospheric implications**
I suggest that you separate Section 4 into '4. Atmospheric implications' where you discuss more carefully the implications citing appropriate literature and 'Section 5. Summary and conclusions' where your study and the main conclusions for atmospheric implications are briefly summarized. The current, extended section 4 is too unstructured and it is not clear what it is a conclusion from your study vs previous studies.

The conclusions on the ranking of 3C* vs OH reactions depend on the concentrations used in experiments or in the atmosphere, respectively. Therefore, please comment on the radical concentrations in your study. If you were unable to retrieve them, please add a general discussion on OH and *C3 concentration in atmospheric waters and how to relate your findings to such concentrations.

l. 296: What were the OH and 3C* concentrations in the study by Yu et al. (2016)?

l. 328: Please report the concentrations of 3C*, 1O2 and OH as used in the experiments by Laurentiis et al., 2013 and Misovich et al. 2021 to make this comparison to your study more solid.
l. 778: How do you quantify 'important'? How much mass of light-absorbing material is predicted to be formed from eugenol based on your experiments for atmospheric conditions?

l. 781 – 784: Small dicarboxylic acids are usually present to a large extent in particles and only evaporate to a small extent.

l. 797-806: The aqueous phase yields cannot be directly compared to gas phase yields because you have to calculate the fraction of the precursor that is present in either phase and scale the yields by them.

**3) Previous work**
l. 599: Huang et al., 2018 is not the only study that reported aqSOA yields from phenolic compounds. There are several more, e.g. (Ma et al., 2021; Smith et al., 2014; Sun et al., 2010). Are the reported values similar? Please discuss.

**4) Technical comments**

l. 113: please write it 'eugenol (allyl guiacol)'

l. 117: 1) 'beech stove' does not seem right here. Do you mean biomass burning of beech wood. ; 2) the study by Liu et al., 2019 does not report on new measurements of emissions; the only cite Bari et al.. Please remove Liu et al., 2019 here.

l. 119: replace 'compounds' by 'compound'

l. 123: 'aerosol mass spectrometer' should be 'aerosol mass spectrometry' to be consistent with the preceding list

l. 175: it should read 'experiments were ...'

l. 193: 'Jenal' should be 'Jena'

l. 221: Is 'some time' necessary here? Either remove 'some time' or quantify the time period.

l. 228: replace 'for distinguish from At' by 'as blank value'.

l. 279: Define BDE here

l. 286-88: This sentence does not read well. Replace by 'When the photon energy is higher than the bond dissociation energy, chemical bonds can break, leading to 287 decomposition of compounds and possibly further mineralization.'

l. 296: remove 'A' at the beginning of the sentence.

l. 302-4: In the headers you use 'photooxidation' while in the subsequent text, you use 'photodegradation'. Unless you mean two different processes here, I suggest using consistent terminology throughout the manuscript.

l. 311: Do you mean 'Excess concentrations' instead of 'Above concentrations'?

l. 332ff: Write (k-kTMP)/k as an equation (as Equation (1) in l. 187) and define k and kTMP. Explain 'contribution' here – contribution to what?

l. 339: either 'One should be cautious to apply' or 'It should be cautioned to apply'

l. 342: replace 'by' by 'be'

l. 375-7: This sentence is not clear. Do you mean 'Since H2O2 was photolyzed at wavelength <300 nm to generate OH radicals, irradiation above 300 nm did not affect the reaction.'?

l. 381: instead of 'shown later', refer to the respective section

l. 391: Add 'conditions' (O2-saturated and N2-saturated conditions...)

l. 394: Either write OH or ·OH. Both is ok but should be used consistently throughout the text.

Figure 3 is of very low quality. Please provide a figure with higher resolution where the axis labels are clearer and less blurry.

l. 445: replace 'direct' by 'photolysis'

l. 468: What do you mean by 'over photoreaction'? 'during the photooxidation'?

l. 468: replace 'slight' by 'light'

l. 524/5: This sentence is not correct, both grammatically and scientifically. There are numerous oligomers from small organic compounds that do not have any fluorescent properties. Either delete this sentence or specify what oligomers you refer to.

l. 536: What do you mean by 'small organic acid'? Small organic acids are usually referred to as acids with a few (1, 2, ..) carbon atoms. Is this what you mean?

l. 537: Is this 'phenomenon' really 'unexpected'? It is well known that the organic content of ambient aerosol particles is composed of 1000s compounds from many different emission sources, gas phase reactions and condensed phase reactions. Thus, the fact that you do not see one particular peak in a

spectrum upon the reaction of a single precursor is by no means 'unexpected'. The reference to Xie et al. 2016 seems therefore random and not appropriate in this context.

l. 204 and l. 548: The header text is basically the same whereas the former is a method and the latter reports results. Please refine the headers such that they differ and appropriately describe the section content.

l. 587: replace 'yield' by 'yields', and 'value' by 'values'

l. 592: replace 'reason' by 'reasons'

l. 623: 'As we known' is grammatically wrong and seems redundant here. Replace by 'The'.

l. 632: Replace this sentence by 'The final f43 values were almost the same as compared to the initial solution'.

l. 660: replace by 'based spectra from the NIST database [add reference!] and on the reactants and reaction conditions' – please check if the content is correct

l. 662: replace 'dominate' by 'dominates' and complete the sentence. What is less important? '..functionalization dominates as compared to...'?

l. 671: What does this sentence refer to? Is it a footnote for the table, or where were eugenol and DMB also shown?

l. 745: 'weaker' than what? Why is the weaker correlation unexpected?

l. 678: replace 'undergo' by 'undergoes'

Ma, L., Guzman, C., Niedek, C., Tran, T., Zhang, Q. and Anastasio, C.: Kinetics and Mass Yields of Aqueous Secondary Organic Aerosol from Highly Substituted Phenols Reacting with a Triplet Excited State, Environ. Sci. Technol., 55(9), 5772–5781, doi:10.1021/acs.est.1c00575, 2021.
Smith, J. D., Sio, V., Yu, L., Zhang, Q. and Anastasio, C.: Secondary Organic Aerosol Production from Aqueous Reactions of Atmospheric Phenols with an Organic Triplet Excited State, Environ. Sci. Technol., 48(2), 1049–1057, doi:10.1021/es4045715, 2014.
Sun, Y., Zhang, Q., Anastasio, C. and Sun, J.: Insights into secondary organic aerosol formed via aqueous-phase reactions of phenolic compounds based on high resolution mass spectrometry, Atmos. Chem. Phys., 10, 4809–4822, 2010.

---

## Author Response (AR3)

Dear Editor

Thanks very much for your comments regarding our manuscript. According to your suggestions, we have improved our manuscript by adding a separate section of "Atmospheric implications" to discuss the findings of this work, and re-wrote the whole conclusions. In addition, we have carefully read the manuscript and performed a thorough language check and made very substantial text changes to eliminate grammar mistakes and broken sentences (please see the manuscript with tracked-changes, and those you suggest in the technical comments below were marked yellow for reference). In particular, we have followed your suggestion to make the terminology consistently regarding the HULIS, aqSOA and oligomers (please also see the replies below). We sincerely hope that the new version now fits the high standards of ACP. Replies to your specific comments are listed below (with your comments repeated in *Italics*):

*1) HULIS vs aqSOA vs oligomers*

*As I pointed out previously, not all aqSOA is HULIS and not all HULIS is aqSOA. Therefore, it cannot be used interchangeably. In addition also oligomers and brown carbon are separate products. While there might be overlaps between these compounds groups (aqSOA, HULIS, BrC, oligomers) they all different definitions and should not be used equally.*

*To clarify and simplify the terminology throughout the paper, I suggest using consistently 'products', 'reaction products' or 'aqueous-phase products'.*

**Reply:** Thanks for pointing out this. We have carefully revised the paper to clarify the usage of these three terms, and followed your suggestions to use 'products', 'reaction products' or 'aqueous-phase products' instead throughout the manuscript.

*l. 94: Did all three studies cited here indeed claim that the investigated reactions yield HULIS? Unfortunately, I do not have access to all of them; however, based on the abstracts, it seems to me that Smith et al., 2016 and Li et al., 2021 only refer to aqSOA or other products. Please verify the terminology in the original literature.*

**Reply:** Yes, the HULIS and aqSOA are used inapporiately previously. We checked the literatures and Li et al. 2021 and Smith et al., 2016 did not direct refers to HULIS. The sentence is now changed to "Tang et al. (2020) reported that aqueous photooxidation of vanillic acid could be a potential source of HULIS."

*l. 30 vs l. 247 vs l. 601 vs l. 633: While in the abstract you call it HULIS, in the experimental section, you call the products sometimes aqSOA, and sometimes HULIS. Please use consistently only one term (preferably 'reaction products') to avoid confusion.*

**Reply:** We now use the term "reaction products" consistently to avoid confusion as suggested.

*l. 437f: Here you refer to studies on HULIS in ambient particulate matter. The HULIS found in these samples are not (necessarily) originating from reactions as investigated in the current study. Therefore, your products (aqSOA) are not comparable to these ambient HULIS.*

**Reply:** Agreed. This sentence and relevant literatures are now deleted.

*l. 482: 'High molecular weight and conjugated structures' are also present in oligomers. Not all oligomers are HULIS. However, if these oligomers are formed by aqueous phase reactions, they can be termed aqSOA.*

**Reply:** Agree. We have re-phrase this sentence, "The enhancement at 300-400 nm may point to products with high molecular weights and conjugated structures, likely relevant with HULIS or oligomers."

*l. 550: Here you could simply delete 'of HULIS'.*

**Reply:** Done

*l. 554 – 560: This text is out of place here and would be redundant if you use 'products'. Also it incorrectly cites the review by Graber and Rudich. 'HULIS' is the acronym for 'HUmic-LIke Substances'. Therefore, the sentence 'So, we inferred most HULIS in this paper was humic-like substance rather than fulvic-like substance' does not make sense.*
*Graber and Rudich discuss that "HULIS relate to the water soluble fraction, which would include only the fulvic acid fraction of humic substances, and exclude the humic acid (base-soluble) and humin (insoluble) fractions of humic substances."*

**Reply:** Agreed. The statement and interpretation with Graber and Rudich (2006) are indeed not appropriate. These text is now deleted.

*l. 596: not all light-absorbing compounds are HULIS; and not all HULIS are light absorbing*

**Reply:** Agreed. We have removed "HULIS".

*l. 771: HULIS and brown carbon are not always the same.*

**Reply:** Yes. "HULIS" is deleted.

**2) Atmospheric implications**

*I suggest that you separate Section 4 into '4. Atmospheric implications' where you discuss more carefully the implications citing appropriate literature and 'Section 5. Summary and conclusions' where your study and the main conclusions for atmospheric implications are briefly summarized. The current, extended section 4 is too unstructured and it is not clear what it is a conclusion from your study vs previous studies.*

**Reply:** We have added a new individual section, "4. Atmospheric implications" as you suggested. Please check the new manuscript.

*The conclusions on the ranking of 3C* vs OH reactions depend on the concentrations used in experiments or in the atmosphere, respectively. Therefore, please comment on the radical concentrations in your study. If you were unable to retrieve them, please add a general discussion on OH and *C3 concentration in atmospheric waters and how to relate your findings to such concentrations.*

**Reply:Yes. This is now discussed in the new section, "**Our study here used 300 μM $H_2O_2$ and 15 μM DMB as sources of OH and $^3C*$, and $^3C*$-mediated oxidation appeared to be faster than OH-initiated oxidation of eugenol. Of course, whether or not $^3C*$ is more important than OH in real atmosphere depends upon the concentrations of OH and $^3C*$. Herrmann et al.

(2010) estimated an average OH level of $0.35 \times 10^{-14}$ M in urban fog water; Kaur and Anastasio (2018) measured $^3C*$ concentration to be $(0.70-15) \times 10^{-14}$ M, 10-100 times higher than OH in ambient fog waters; Kaur et al. (2019) determined both OH and $^3C*$ concentrations in PM extracts, OH steady-state concentration was $4.4(\pm2.3) \times 10^{-16}$ M, similar to its level in fog, cloud and rain, while $^3C*$ concentration was $1.0(\pm0.4) \times 10^{-13}$ M, a few hundred times higher than OH and nearly double its average value in fog. Therefore, together with these measurements, our findings indicate a likely more important role of $^3C*$ than OH in aqueous-phase (especially aerosol water) reactions. In addition, quenching experiments reveal that $O_2$ can inhibit eugenol degradation by effectively scavenging $^3C*$ radical while it can promote degradation by fostering radical chain reactions in OH-induced oxidation, which offer insights to control of reaction pathways by regulating the ROS generations; of course, such operation requires application of highly sensitive EPR method. **"**

*l. 296: What were the OH and 3C* concentrations in the study by Yu et al. (2016)?*
**Reply:** The initial $H_2O_2$ and DMB concentrations to generate OH and $^3C*$ are 100 μM and 5 μM, respectively, which are the same ratio as those used in this work 300 μM $H_2O_2$ and 15 μM DMB). This information was added in the text now.

*l. 328: Please report the concentrations of 3C*, 1O2 and OH as used in the experiments by Laurentiis et al., 2013 and Misovich et al. 2021 to make this comparison to your study more solid.*
**Reply:** Thanks. We have re-wrote the sentences, "Previously, Laurentiis et al. (2013) reported that 4-carboxybenzophenone (70 μM) can act as $^3C^*$ and the photosensitized degradation was more effective than oxidants such as OH, $O_3$, etc. Misovich et al. (2021) investigated the aqueous DMB-photosensitized reaction (5 μM, same concentration as this study) also demonstrated that $^3C^*$ was the greatest contributor to phenol or guaiacyl acetone degradation, followed by $^1O_2$, while both OH and $^1O_2$ contributions were relatively minor."

*l. 778: How do you quantify 'important'? How much mass of light-absorbing material is predicted to be formed from eugenol based on your experiments for atmospheric conditions?*
**Reply:** We now deleted "important"

*l. 781 – 784: Small dicarboxylic acids are usually present to a large extent in particles and only evaporate to a small extent.*
**Reply:** Agree. This sentence seems to be incorrect, at least for dicarboxylic acids. It is now deleted.

*l. 797-806: The aqueous phase yields cannot be directly compared to gas phase yields because you have to calculate the fraction of the precursor that is present in either phase and scale the yields by them.*
**Reply:** Agree. The arguments are now modified per your suggestion, and is incorporated into the new "Atmospheric implications" section

**3) Previous work**
*l. 599: Huang et al., 2018 is not the only study that reported aqSOA yields from phenolic*

compounds. There are several more, e.g. (Ma et al., 2021; Smith et al., 2014; Sun et al., 2010). Are the reported values similar? Please discuss.

**Reply**:Yes. We have now cited more literatures, and discuss them accordingly. "The product mass yields obtained in this work overall agree with those reported previously for phenolic compounds. For examples, Huang et al. (2018) reported mass yields of 30-120% for syringaldehyde and acetosyringone; Smith et al. (2014) found that mass yields of aqSOA from three phenols with $^3C*$ were nearly 100%, and Ma et al. (2021) reported a yield ranging from 59 to 99% for six highly substituted phenols with $^3C*$; Mass yields of SOA from three benzene-diols were near 100% with both OH and $^3C*$ oxidants, as reported in Smith et al. (2015); Direct photolysis of phenolic carbonyls, and oxidation of syringol by $^3C*$, had SOA mass yields ranging from 80 to 140% (Smith et al., 2016)"

**4) Technical comments**
*l. 113: please write it 'eugenol (allyl guiacol)'*
**Reply**:Done

*l. 117: 1) 'beech stove' does not seem right here. Do you mean biomass burning of beech wood. ; 2) the study by Liu et al., 2019 does not report on new measurements of emissions; the only cite Bari et al.. Please remove Liu et al., 2019 here.*
**Reply**:1) Yes, we changed to "beech wood burning"; 2) Liu et al., 2019 removed.

*l. 119: replace 'compounds' by 'compound'*
*l. 123: 'aerosol mass spectrometer' should be 'aerosol mass spectrometry' to be consistent with the preceding list*
*l. 175: it should read 'experiments were …'*
*l. 193: 'Jenal' should be 'Jena'*
**Reply**:Done

*l. 221: Is 'some time' necessary here? Either remove 'some time' or quantify the time period.*
**Reply**:We removed "some time"

*l. 228: replace 'for distinguish from At' by 'as blank value'.*
**Reply**:**Done**

*l. 279: Define BDE here*
**Reply**:It should be "bond dissociation energies (BDEs)"

*l. 286-88: This sentence does not read well. Replace by 'When the photon energy is higher than the bond dissociation energy, chemical bonds can break, leading to 287 decomposition of compounds and possibly further mineralization.'*
*l. 296: remove 'A' at the beginning of the sentence.*
**Reply**:Done

*l. 302-4: In the headers you use 'photooxidation' while in the subsequent text, you use*

*'photodegradation'. Unless you mean two different processes here, I suggest using consistent terminology throughout the manuscript.*

**Reply:** Thanks. We use "photooxidation" consistently throughout the manuscript now.

*l. 311: Do you mean 'Excess concentrations' instead of 'Above concentrations'?*

**Reply:** Yes

*l. 332ff: Write (k-kTMP)/k as an equation (as Equation (1) in l. 187) and define k and kTMP. Explain 'contribution' here – contribution to what?*

**Reply:** Sorry this is not made clear previously. Now a short paragraph is added in the revised manuscript, "We propose to use the following Eq.(5) to estimate the contribution of a certain ROS ($Ct_{ROS}$) to eugenol degradation:

$$Ct_{ROS}=k_{ROS}/k=(k-k_{quencher})/k \quad\quad\quad (5)$$

Here $k_{ROS}$ is the rate constant contributed by the ROS, which is defined as the difference between the original rate constant in $^3C*$-initiated oxidation (k) and the rate constant ($k_{quencher}$) after the target ROS has been completely scavenged by its corresponding quencher. k and $k_{quencher}$ in fact refer to those reported in Fig. S1b. "

*l. 339: either 'One should be cautious to apply' or 'It should be cautioned to apply'*
*l. 342: replace 'by' by 'be'*

**Reply:** Done

*l. 375-7: This sentence is not clear. Do you mean 'Since H2O2 was photolyzed at wavelength <300 nm to generate OH radicals, irradiation above 300 nm did not affect the reaction.'?*

**Reply:** Yes, we have re-written this sentence as suggested.

*l. 381: instead of 'shown later', refer to the respective section*

**Reply:** Yes, it refers to Section 3.2.3, it is now made clear.

*l. 391: Add 'conditions' (O2-saturated and N2-saturated conditions...)*

**Reply: Done**

*l. 394: Either write OH or ·OH. Both is ok but should be used consistently throughout the text.*

**Reply:** We now use OH consistently throughout the text.

*Figure 3 is of very low quality. Please provide a figure with higher resolution where the axis labels are clearer and less blurry.*

**Reply:** Figure 3 is replaced with larger and higher resolution ones now.

*l. 445: replace 'direct' by 'photolysis'*

**Reply:** Done

*l. 468: What do you mean by 'over photoreaction'? 'during the photooxidation'?*

**Reply:** Yes

*l. 468: replace 'slight' by 'light'*
**Reply:** Done

*l. 524/5: This sentence is not correct, both grammatically and scientifically. There are numerous oligomers from small organic compounds that do not have any fluorescent properties. Either delete this sentence or specify what oligomers you refer to.*
**Reply:** This sentence is deleted now.

*l. 536: What do you mean by 'small organic acid'? Small organic acids are usually referred to as acids with a few (1, 2, ..) carbon atoms. Is this what you mean?*
**Reply:** Yes, we have now made it clear.

*l. 537: Is this 'phenomenon' really 'unexpected'? It is well known that the organic content of ambient aerosol particles is composed of 1000s compounds from many different emission sources, gas phase reactions and condensed phase reactions. Thus, the fact that you do not see one particular peak in a spectrum upon the reaction of a single precursor is by no means 'unexpected'. The reference to Xie et al. 2016 seems therefore random and not appropriate in this context.*
**Reply:** Agree. This sentence and the later sentence citing Xie et al., 2016 were both deleted.

*l. 204 and l. 548: The header text is basically the same whereas the former is a method and the latter reports results. Please refine the headers such that they differ and appropriately describe the section content.*
**Reply:** The later header text is now changed to "Characteristics of HULIS"

*l. 587: replace 'yield' by 'yields', and 'value' by 'values'*
*l. 592: replace 'reason' by 'reasons'*
*l. 623: 'As we known' is grammatically wrong and seems redundant here. Replace by 'The'.*
*l. 632: Replace this sentence by 'The final f43 values were almost the same as compared to the initial solution'.*
**Reply:** Done

*l. 660: replace by 'based spectra from the NIST database [add reference!] and on the reactants and reaction conditions' – please check if the content is correct*
**Reply:** It is correct. We have changed the sentence, and add a reference.

*l. 662: replace 'dominate' by 'dominates' and complete the sentence. What is less important? '..functionalization dominates as compared to...'?*
**Reply:** The sentence is re-phrased, "We also found 4-(1-hydroxypropyl)-2-methoxyphenol (product 8) was relatively abundant (Fig.S7), suggesting functionalization might be dominant as compared to oligomerization and fragmentation."

*l. 671: What does this sentence refer to? Is it a footnote for the table, or where were eugenol and DMB also shown?*

**Reply:** It is a footnote for the table, and refers to the "name" of identified compounds. We want to clarify that precursor eugenol and DMB are also listed in the table.

*l. 745: 'weaker' than what? Why is the weaker correlation unexpected?*
**Reply:** We changed to "weak", and deleted "unexpected", which is not appropriate.

*l. 678: replace 'undergo' by 'undergoes'*
**Reply:** Done

---

## Editor Decision (ED3)

**I. Major comments**

**1) Atmospheric implications**
The stated implications exceed the conclusions that can be drawn on the current study

**l. 42/43:** "as well as its impacts on particulate matter concentration and toxicity, radiative balance and climate change" – Since these aspects are not discussed in the paper, it seems that these conclusions are too far reaching. – I suggest deleting.

**l. 765 – 767:** The paper by Ma et al., 2021 does not quantify the amount of phenols into the aqueous phase. I do not think that your statement is correct; there are several issues:
1) Syringol and eugenol are equally substituted phenols, as they have both three functional groups on the aromatic ring.
2) How can such different partitioning behavior be explained? The Henry's law constants for eugenol and syringol are 729 M/atm (Sander, 2015; doi:10.5194/acp-15-4399-2015) and 2.6e4 M/atm (www.atmos-chem-phys-discuss.net/10/C2298/2010/).
Thus, one would expect syringol to partition ~30 times more efficiently into water.
3) The Henry's law constants allow an estimate of the partitioned fraction into an aqueous phase by appropriate unit conversion of the concentration ratio as given by Henry's law, e.g:

$$CR\ [dimensionless] = \ K_H \left[\frac{mol}{L(aq)\ atm}\right] LWC \underbrace{\left[\frac{L(aq)}{cm^3(gas)}\right] \frac{N_A \left[\frac{molecules}{mol}\right]}{2.5e19 \left[\frac{molecules}{cm^3(g)\ atm}\right]}}_{Conversion\ factor\ 'X'}$$

For a typical cloud water content of 0.3 g/m3 (=3e-10 L(aq)/cm3(gas)): X = 7.22e-6; for typical aerosol water contents of 50 μg/m³, X = 1.2e-10

The fraction in the aqueous phase can then be calculated as
$$\varepsilon_{aq} = \frac{CR}{CR+1}$$
Thus, the fraction partitioned into the aqueous phase ($\varepsilon_{aq}$) for compounds of different Henry's law constants can be calculated for clouds and fogs

|  | LWC = 0.3 g/m³ | LWC = 50 μg/m³ |
|---|---|---|
| KH = 729 M/atm | $\varepsilon_{aq}$ = 0.005 | $8.7 \cdot 10^{-8}$ |
| KH = 2.6e4 M/atm | $\varepsilon_{aq}$ = 0.158 | $3.2 \cdot 10^{-6}$ |

As indicated in this interactive comment (https://acp.copernicus.org/preprints/10/C2298/2010/acpd-10-C2298-2010-print.pdf - citing a reference that is not open access) , aqueous phase concentrations of methoxy phenols may be higher by a factor of 3-4 than predicted by Henry's law. Even such enhancement would not lead to a significant partitioning of eugenol or other methoxy phenols into aerosol water (<<1%).
Reasons why reactions in aerosol water may be more important than in cloud water might include different reaction pathways due to concentration effects such as oligomerization and/or enhanced

solubility due to ionic strength effects ('salting-in'). However, phenols show generally a salting-out effect (Wang et al., Envoron Sci. Technol., 2014 https://doi.org/10.1021/es5035602)
Unless you can argue that such effects occur for eugenol in aerosol water as compared to cloud water, your discussion on the relevance in aerosol water is not convincing.

**l. 780 - 784:** The liquid water content of aerosol water is about 10000 times smaller than that of cloud (~50 $\mu g$ m-3 vs 0.5 g m-3). Even if the reaction rates in aerosol water were 10 times higher than those in cloud water due to higher oxidant concentration, the overall importance of the aerosol phase would be still 100 times smaller than that of chemical reactions in cloud water. Such values should be taken into account in the discussion.

**l. 813 – 816:** '...our findings here underscore the potential of aqueous processing on 813 the enhancement of particle toxicity. Considering high PM concentration is often accompanied with cold and humid weather conditions, the additional adverse health effects caused by aqueous oxidation may amplify the health hazards of PM pollution.'
These conclusions are quite far-fetched. 1) If you do not compare the OP from gas-phase reactions, you cannot state that aqueous phase reactions cause a higher oxidation potential'. 2) Toxicity is not only defined by OP.

**l. 821:** You only investigated one aspect of potential adverse health effects, i.e. the oxidation potential. This is by no means a 'systematic investigation of toxicity'.

**l. 851:** '...our findings highlight the importance of aqueous oxidation of BB emissions to SOA formation, its potentially important role in affecting radiative balance and climate through formation of BrC, as well as possible additional adverse health effects. Such effects should be considered in air quality or climate models to better assess the influence of BB emissions.'
Again, these conclusions are way overstating your findings. You neither reported absorption coefficients, nor any radiation calculation, nor the yields of brown carbon nor an estimate of the amount of SOA that can be formed from such precursors.

**2) Wrong terminology**
l. 121, l. 196, l. 574, l. 819: 3C* is not a radical. Write '...and oxidation by OH radicals and 3C* triplet states' – please check the remainder of the manuscript for other instances of its wrong terminology

Sections 3.4 and 3.5: I am still confused by your use of 'reaction products' and 'HULIS'. In Section 3.4, you describe HULIS properties, whereas in Section 3.5 you talk about Reaction products. Are the HULIS you are referring to the same or just a subset of the total products? – I realized that you do mention it briefly in l. 725 – but this assumption should be added earlier in the text.

l. 459/460: 'which can be explained by the transfer of electrons from 3C* to O2 to form 1O2' – this is chemically wrong. Both O2 and 1O2 have the same number of electrons. Please clarify what you mean here.

l. 826: 'Photolysis rate constants' only refers to the direct photolysis, not to the oxidation by OH. Please correct.

**3) Methodology**
l. 316 - 320: 'The optimum molar ratio of eugenol to quencher was chosen when the inhibition degree of eugenol degradation unchanged with the increase of added quencher' does not read well. I do not understand what you mean by 'inhibition degree' or 'inhibitory degree' in this context – they usually refer to enzyme reactions.
Do you mean 'The optimum molar ratio of eugenol to quencher was selected such that the eugenol degradation did not change with the increase of added quencher'?

l. 335 - 342: I do not understand this text. What contribution is calculated by Equation 5?
Does it only refer to the reactions of 3C*? What units does k and kROS have?
It is not clear how the various contributions can add up to more than 100%.
I understand it as follows: In the course of the reaction of the triplet state, other reactive species such as OH, O2- and 1O2 are formed. They all react with eugenol and thus contribute to its degradation. Thus, there are four contributions (3C*, OH, 1O2 and O2-) that cause the concentration of eugenol to decrease. If the full decrease were normalized to 100%, the sum of the four contributions should add up to these 100% - not more. Can you express your results in this sense? Or did I misunderstand Equation 5? If so, please clarify.

**Figure quality:**
Figure 3: I do not see much difference in this figure compared to the previous version. The insets still look blurry and are hard to read. I suggest moving them outside of the figures and making separate panels, the same size as the three panels a, b, c.

Figure 5: Please make the legend in panel a) larger or even move it out of the panels so that you can use a readable font size.

Captions of Figures 7 and 8: Please give more details in this figure caption. The legend of the figures shows 'direct', OH, and 3C* - which are not conditions but refer to reaction pathways. The reader has to be able to understand what 'conditions' refers to.

Figure 9: I do not understand the figure caption and the legend. The caption states (a) direct photolysis, (b) OH-initiated reaction, (c) 3C* initiated oxidation.
However, the legend in panel a) implies that all three processes are shown in panels a-c
Please also add a caption for panels d-f.

Scheme 1: I see only one text in red here: MW 164 Eugenol; all other text looks pink to me. It might be clearer to just say: "The compounds labeled by Product 1-9 are those identified by GC-MS (Table 2)."

**II. Technical comments**

l. 1: add 'and' – 'Optical and chemical properties ...'

l. 16: remove 'etc' – it is very vague and not very powerful as a first sentence of an abstract.

l. 21 – 23: Quenching experiments verified that 3C* indeed played a dominant role in 3C*-initiated oxidation, while O2•- generated was important for OH- initiated oxidation.

This sentence does not read well. It seems obvious that *C3 plays a dominant role in *C3 reactions. What do you mean by the second part of the sentence? - Do you imply that O2- is a main reactant with eugenol? Please clarify.

l. 81: remove 'as'

l. 96: add 'which' : '..which had some similarities'

l. 120: remove 'too'
l. 125: Which ROS do you refer here to? You only compare OH and 3C* reactions – OH is an ROS, 3C* is not.

l. 149: add 'the' : '...at the bottom'

l. 153: add 'the': '...at the surface'

l. 165: remove 'the' before 'aluminum foil'

l. 209: replace 'blew' by 'blown'

l. 238: replace 'Products' by 'Product'

l. 258: Are there any other major factors beyond atomization efficiency and carrier gas flow? If so, list them, if not, remove 'etc'.

l. 282: 'The lowest BDE was found for the O-H bond and C=C bond.' Is this a finding from your study or just repeating the information of the previous sentence? If the latter, it can be removed.

l. 290: in lines 280-282, you state that all BDEs are in the range of 340 – 403 kJ/mol. Thus, the photon energy of 412 kJ/mol should be sufficient to break any of these bonds. Please clarify.

l. 301: 'Regressed' is not an English word to be used in this context. Change to 'first-order rate constants obtained based on Equation 1'

l. 331: What other oxidants are you referring to here? If none, remove 'etc'

l. 332: remove 'it'

l. 353: do you mean 'quenching effect' rather than 'inhibitive effect'?

l. 443: Replace 'dramatically' by 'quickly' or 'significantly'

l. 447: replace 'likely ascribing' by 'which can be likely ascribed'

l. 447-449: 'Note a small amount of acids can  change solution pH significantly when original pH is high, but cannot change pH remarkably when the original solution pH was low.'
This sentence does not seem necessary. Instead, I suggest starting the next sentence as 'Since the solution was acidic (pH = 3), we cannot rule out ...'

l. 465: replace 'dissolve' by 'dissolved'

l. 498: What does the addition of 'possibly linking with HULIS or oligomers' add to the content here?

l. 526: Please add a reference for the statement that HULIS are highly oxygenated.

l. 536: replace 'caveats' by 'caution'

l. 537: what do you mean by 'complicated'? I suggest removing.

l. 544- 546: ' EEM fluorescence spectra of HULIS from fog water are reported to have peaks at shorter excitation and emission wavelengths than those of terrestrial fulvic acids  (Graber and Rudich, 2006).' – This sentence seems out of place here. What does it contribute to the discussion here?

l. 549: Can you quantify 'a few' in this context?

l. 579: This still adds to the confusion I had pointed out in the last round of reviews: Are you implying that HULIS cannot be high MW oligomers?

l. 610: 'simulated lights' is not correct. Do you mean 'light intensity' or 'wavelength' – or both?

l. 641/2: 'As a result, all data points located outside the f44 vs. f43 space established by Ng et al. (2010) for ambient aerosols, owing to the relatively low f43 values.'
This sentence seems grammatically wrong; it is also not clear what you try to say.

l. 725: Can you quantify the fraction of HULIS to total reaction products?

l. 750: 'This finding further indicates the effectiveness of DTT method to represent OP.' Which finding do you refer to here? Isn't this a circular and redundant statement? – The DTT method was developed to quantify OP – thus, it seems obvious that it can used to do this.

l. 791: replace 'photolyze itself' by 'undergo direct photolysis'

l. 800: replace 'foil' by 'fuel'

l. 801 – 803: 'Aqueous oxidation of 4-nitrophenol with OH can lead to a photobleaching effect too.' Nitrophenols are not comparable at all to the compounds you studied. There, the nitro group causes light absorption – thus this comparison seems out pf place and not relevant.

**Data availability:** Please deposit your data in a suitable repository according to the journal data policy https://www.atmospheric-chemistry-and-physics.net/policies/data_policy.html

---

## Author Response (AR4)

**Response to comments**

Dear Editor

Thank you very much again for your comments regarding our manuscript! According to your guidance, we do realize that some implications drawn from our findings are overstated and may not be appropriate, they were either deleted or modified. In addition, other comments were also addressed point-by-point (as shown below) with your comments repeating in *italics*. All relevant data is now deposited on a website that is specified in the Data availability section. The manuscript was also checked carefully again with all your technical corrections incorporated, please check the manuscript with tracked changes.

*I. Major comments*
*1) Atmospheric implications*
*The stated implications exceed the conclusions that can be drawn on the current study*
**Reply:** Agree. We have either deleted or modified some statements, please see below.

*I. 42/43: "as well as its impacts on particulate matter concentration and toxicity, radiative balance and climate change" – Since these aspects are not discussed in the paper, it seems that these conclusions are too far reaching. – I suggest deleting.*
**Reply:** Agree. The sentence is deleted.

*I. 765 – 767: The paper by Ma et al., 2021 does not quantify the amount of phenols into the aqueous phase. I do not think that your statement is correct; there are several issues:*
*1) Syringol and eugenol are equally substituted phenols, as they have both three functional groups on the aromatic ring.*
*2) How can such different partitioning behavior be explained? The Henry's law constants for eugenol and syringol are 729 M/atm (Sander, 2015; doi:10.5194/acp-15-4399-2015) and 2.6e4 M/atm (www.atmos-chem-phys-discuss.net/10/C2298/2010/). Thus, one would expect syringol to partition ~30 times more efficiently into water.*
*3) The Henry's law constants allow an estimate of the partitioned fraction into an aqueous phase by appropriate unit conversion of the concentration ratio as given by Henry's law, e.g:*

$$CR\,[dimensionless] = K_H \left[\frac{mol}{L(aq)\,atm}\right] LWC \left[\frac{L(aq)}{cm^3(gas)}\right] \underbrace{\frac{N_A \left[\frac{molecules}{mol}\right]}{2.5e19 \left[\frac{molecules}{cm^3(g)\,atm}\right]}}_{Conversion\ factor\ 'X'}$$

*For a typical cloud water content of 0.3 g/m3 (=3e-10 L(aq)/cm3(gas)): X = 7.22e-6; for typical aerosol water contents of 50 ug/m3, X = 1.2e-10*
*The fraction in the aqueous phase can then be calculated as*

$$\varepsilon_{aq} = \frac{CR}{CR+1}$$

*As indicated in this interactive comment*
*(https://acp.copernicus.org/preprints/10/C2298/2010/acpd-10-C2298-2010-print.pdf        -*

*citing a reference that is not open access), aqueous phase concentrations of methoxy phenols may be higher by a factor of 3-4 than predicted by Henry's law. Even such enhancement would not lead to a significant partitioning of eugenol or other methoxy phenols into aerosol water (<<1%). Reasons why reactions in aerosol water may be more important than in cloud water might include different reaction pathways due to concentration effects such as oligomerization and/or enhanced solubility due to ionic strength effects ('salting-in'). However, phenols show generally a salting-out effect (Wang et al., Envoron Sci. Technol., 2014 https://doi.org/10.1021/es5035602) Unless you can argue that such effects occur for eugenol in aerosol water as compared to cloud water, your discussion on the relevance in aerosol water is not convincing.*

**Reply:** Thanks very much for your clarification and reminder! We read it carefully and agree with your viewpoint. Ma et al. (2021) does state that "The low liquid water content (LWC) of ALW leads to very limited partitioning of simple phenols to particle water; e.g., less than 0.001% of syringol will partition into the water phase for an ALW content of 100 µg m$^{-3}$," and they argued that for the six phenols they studied, the aqueous fractions can be 2-58%, therefore highlighted the importance of their precursors in aerosol water reactions. However, this may not be applicable to eugenol we used here, as its Henry's law coefficient is not high and eugenol is not a highly substituted phenol as ones in that paper. We are sorry that we just blindly cited their reference without considering the difference between the henry's law constant of eugenol and those in Ma et al. (2021); eugenol indeed is unable to partition into aqueous water as your calculations have verified, and the importance of aerosol water reactions of eugenol is small. Even though the reactions of some other high substituted phenols might be important in aerosol water but this is not a convincing statement that can be drawn base on our current study on eugenol. Therefore such statement is now deleted.

*l. 780 - 784: The liquid water content of aerosol water is about 10000 times smaller than that of cloud (~50 ug m-3 vs 0.5 g m-3). Even if the reaction rates in aerosol water were 10 times higher than those in cloud water due to higher oxidant concentration, the overall importance of the aerosol phase would be still 100 times smaller than that of chemical reactions in cloud water. Such values should be taken into account in the discussion.*

**Reply:** Agree. This point is now added, our intention is to state that $^3$C* may be important than OH in aerosol water (or in cloud water), but not to state that role of $^3$C* (or OH) in aerosol water is more important that it in cloud water. This is now made clear.

*l. 813 – 816: '…our findings here underscore the potential of aqueous processing on 813 the enhancement of particle toxicity. Considering high PM concentration is often accompanied with cold and humid weather conditions, the additional adverse health effects caused by aqueous oxidation may amplify the health hazards of PM pollution.'*
*These conclusions are quite far-fetched. 1) If you do not compare the OP from gas-phase reactions, you cannot state that aqueous phase reactions cause a higher oxidation potential'. 2) Toxicity is not only defined by OP.*

**Reply:** Agree. This is indeed over-interpreted. Without evidence that proves OP of aq-SOA is higher than that of gasSOA, this argument is incorrect, and toxicity is not only indicated by OP. This sentence is now deleted.

*l. 821: You only investigated one aspect of potential adverse health effects, i.e. the oxidation potential. This is by no means a 'systematic investigation of toxicity'.*
**Reply:** Agree. "systamatically" is now deleted.

*l. 851: '…our findings highlight the importance of aqueous oxidation of BB emissions to SOA formation, its potentially important role in affecting radiative balance and climate through formation of BrC, as well as possible additional adverse health effects. Such effects should be considered in air quality or climate models to better assess the influence of BB emissions.'*
*Again, these conclusions are way overstating your findings. You neither reported absorption coefficients, nor any radiation calculation, nor the yields of brown carbon nor an estimate of the amount of SOA that can be formed from such precursors.*
**Reply:** Agree. This is indeed over-interpreted, and is now deleted.

*2) Wrong terminology*
*l. 121, l. 196, l. 574, l. 819: 3C* is not a radical. Write '…and oxidation by OH radicals and 3C* triplet states' – please check the remainder of the manuscript for other instances of its wrong terminology*
**Reply:** Thanks for pointing out this. Wrong use of "radical" for $^3C*$ is now all corrected throughout the manuscript.

*Sections 3.4 and 3.5: I am still confused by your use of 'reaction products' and 'HULIS'. In Section 3.4, you describe HULIS properties, whereas in Section 3.5 you talk about Reaction products. Are the HULIS you are referring to the same or just a subset of the total products? – I realized that you do mention it briefly in l. 725 – but this assumption should be added earlier in the text.*
**Reply:** Yes, HULISs is just a subset of the total products, this is now made clear in the beginning of Section 3.5 as you suggested, "HULIS is only a subset of the products from aqueous oxidation, and here we used AMS to further quantify the total reaction products". Section 3.4 investigated HULIS as EEM measurement suggests the presence of HULIS. Section 3.5 is separated from Section 3.4, and we investigated the total reaction products by using AMS. To avoid confusion, we also deleted the first two sentences referring to HULIS in Section 3.7, as this section investigated OPs of total products not only HULIS.

*l. 459/460: 'which can be explained by the transfer of electrons from 3C* to O2 to form 1O2' – this is chemically wrong. Both O2 and 1O2 have the same number of electrons. Please clarify what you mean here.*
**Reply:** This sentence is now corrected,"The maximum consumed DO was found in $^3C*$-initiated oxidation, which might be explained by the consumption of $O_2$ that reacts with $^3C*$ form $^1O_2$ (R5)."

*l. 826: 'Photolysis rate constants' only refers to the direct photolysis, not to the oxidation by OH. Please correct.*
**Reply:** Thanks. Wrong use of this terminology is now corrected throughout the paper.

*3) Methodology*
*l. 316 - 320: 'The optimum molar ratio of eugenol to quencher was chosen when the inhibition degree of eugenol degradation unchanged with the increase of added quencher' does not read well. I do not understand what you mean by 'inhibition degree' or 'inhibitory degree' in this context – they usually refer to enzyme reactions. Do you mean 'The optimum molar ratio of eugenol to quencher was selected such that the eugenol degradation did not change with the increase of added quencher'?*

**Reply:** The use of "inhibition degree" or "inhibitory degree" is probably not accurate. The sentence should be read as you suggested, it is now changed.

*l. 335 - 342: I do not understand this text. What contribution is calculated by Equation 5? Does it only refer to the reactions of 3C*? What units does k and kROS have? It is not clear how the various contributions can add up to more than 100%. I understand it as follows: In the course of the reaction of the triplet state, other reactive species such as OH, O2- and 1O2 are formed. They all react with eugenol and thus contribute to its degradation. Thus, there are four contributions (3C*, OH, 1O2 and O2-) that cause the concentration of eugenol to decrease. If the full decrease were normalized to 100%, the sum of the four contributions should add up to these 100% - not more. Can you express your results in this sense?*
*Or did I misunderstand Equation 5? If so, please clarify.*

**Reply:** Thanks for your comment. We have explained but not well why the sum of contributions might be larger than 100%. After a careful consideration, it is indeed not proper to call it "contribution" of ROS through equation (5). The calculation is only a reflection of the relative importance of that ROS, not an absolute contribution. The quenching test works as follows: We first determined the original rate constant with $^{3}$C* (or OH) as $k$, then we added a quencher to completely scavenge a certain ROS, and determined a new $k$ as $k_{quencher}$; the reduction of $k$ (equal to ($k$-$k_{quencher}$)) reflects the impact of that ROS on $k$; then ($k$-$k_{quencher}$)/$k$ can be used as an indicator of the relative importance of that ROS to oxidation. However, as we explained, addition of a specific quencher will remove the oxidation by a certain ROS, but on the other hand, the reactions via other ROS may be enhanced if compared to the one without that quencher; in this regard, the sum of calculated reductions (no longer called as "contributions") from all four ROS might not add up to 100%. To answer your question, the equation also refers to quenching experiments for OH-oxidation, and the units of $k$ and $k_{quencher}$ ($k_{ROS}$ is no longer used) are both s$^{-1}$ (the first-order rate constant). This is now made clear in the manuscript.

*Figure quality:*
*Figure 3: I do not see much difference in this figure compared to the previous version. The insets still look blurry and are hard to read. I suggest moving them outside of the figures and making separate panels, the same size as the three panels a, b, c.*

**Reply:** As suggested, the insets are moved outside and named separately as (d-f)

*Figure 5: Please make the legend in panel a) larger or even move it out of the panels so that you can use a readable font size.*

**Reply:** The legend in (a) is now enlarged so it can be clearly seen.

*Captions of Figures 7 and 8: Please give more details in this figure caption. The legend of the figures shows 'direct', OH, and 3C\* - which are not conditions but refer to reaction pathways. The reader has to be able to understand what 'conditions' refers to.*
**Reply:** Thanks. The three conditions are now specified as "under direct photolysis, OH-initiated oxidation, and 3C\*-initiated oxidation" in captions of Figure 7 and 8 (as well as Figure 1 and Figure 4, and relevant texts in the manuscript)

*Figure 9: I do not understand the figure caption and the legend. The caption states (a) direct photolysis, (b) OH-initiated reaction, (c) 3C\* initiated oxidation. However, the legend in panel a) implies that all three processes are shown in panels a-c . Please also add a caption for panels d-f.*
**Reply:** Sorry for the mistake. (a),(b) and (c) before direct photolysis, OH-initiated oxidation and 3C\*-initiated oxidation should be (d), (e) and (f), which refer to the f44 vs f43 plots for the three sets of oxidation experiments.

*Scheme 1: I see only one text in red here: MW 164 Eugenol; all other text looks pink to me. It might be clearer to just say: "The compounds labeled by Product 1-9 are those identified by GC-MS (Table 2)."*
**Reply:** The caption is changed to "The red text represents the precursor, and the compounds labeled by Product 1-9 are those identified by GC-MS (Table 2). "

*II. Technical comments*
*l. 1: add 'and' – 'Optical and chemical properties ...'*
*l. 16: remove 'etc' – it is very vague and not very powerful as a first sentence of an abstract.*
**Reply:** Done.

*l. 21 – 23: Quenching experiments verified that $^{3}C^{*}$ indeed played a dominant role in 3C\*-initiated oxidation, while O2•- generated was important for OH- initiated oxidation.*
*This sentence does not read well. It seems obvious that \*C3 plays a dominant role in \*C3 reactions. What do you mean by the second part of the sentence? - Do you imply that O2- is a main reactant with eugenol? Please clarify.*
**Reply:** Here is what we want to state, "During $^{3}C^{*}$-initiated oxidation, there are different reactive oxygen species (ROS) including $^{3}C^{*}$ OH, $^{1}O_2$ and $O_2^{\cdot -}$ that can participate in oxidation of eugenol, quenching experiments verified $^{3}C^{*}$ was the most important one; while during OH-initiated oxidation, $O_2^{\cdot -}$ was a more important ROS than OH to oxidize eugenol."

*l. 81: remove 'as'*
*l. 96: add 'which' : '..which had some similarities'*
*l. 120: remove 'too'*
**Reply:** Done.

*l. 125: Which ROS do you refer here to? You only compare OH and 3C\* reactions – OH is an ROS, 3C\* is not.*
**Reply:** This sentence is deleted, as the content was in fact included in the previous sentence.

*l. 149: add 'the' : '...at the bottom'*
*l. 153: add 'the': '...at the surface'*
*l. 165: remove 'the' before 'aluminum foil'*
*l. 209: replace 'blew' by 'blown'*
*l. 238: replace 'Products' by 'Product'*
**Reply:** Done.

*l. 258: Are there any other major factors beyond atomization efficiency and carrier gas flow? If so, list them, if not, remove 'etc'.*
**Reply:** "etc" is removed.

*l. 282: 'The lowest BDE was found for the O-H bond and C=C bond.' Is this a finding from your study or just repeating the information of the previous sentence? If the latter, it can be removed.*
**Reply:** It is information from previous literature, and is now removed as suggested.

*l. 290: in lines 280-282, you state that all BDEs are in the range of 340 – 403 kJ/mol. Thus, the photon energy of 412 kJ/mol should be sufficient to break any of these bonds. Please clarify.*
**Reply:** Here what we want to state is that since the energies of 300 nm and 350 nm lights are 412 kJ/mol and 353 kJ/mol, both of these energies are higher than that of the weakest BDE of 340 kJ/mol, therefore both lights can lead to breakage of eugenol. It is changed to, "The energy of photon of 300 nm is 412 kJ/mol therefore is able to break all major bonds in eugenol, and the energy of 350 nm is 352 kJ/mol, being able to break some bonds in eugenol as well. Thus, eugenol can be easily decomposed after absorbing the photons. "

*l. 301: 'Regressed' is not an English word to be used in this context. Change to 'first-order rate constants obtained based on Equation 1'*
*l. 331: What other oxidants are you referring to here? If none, remove 'etc'*
*l. 332: remove 'it*
*l. 353: do you mean 'quenching effect' rather than 'inhibitive effect'?*
*l. 443: Replace 'dramatically' by 'quickly' or 'significantly'*
*l. 447: replace 'likely ascribing' by 'which can be likely ascribed'*
*l. 447-449: 'Note a small amount of acids can change solution pH significantly when original pH is high, but cannot change pH remarkably when the original solution pH was low.'*
*This sentence does not seem necessary. Instead, I suggest starting the next sentence as 'Since the solution was acidic (pH = 3), we cannot rule out ...'*
*l. 465: replace 'dissolve' by 'dissolved'*
**Reply:** Done.

*l. 498: What does the addition of 'possibly linking with HULIS or oligomers' add to the content here?*
**Reply:** It is now deleted.

*l. 526: Please add a reference for the statement that HULIS are highly oxygenated.*
**Reply:** "highly oxygenated" should not be used here.

*l. 536: replace 'caveats' by 'caution'*
*l. 537: what do you mean by 'complicated'? I suggest removing.*
*l. 544- 546: ' EEM fluorescence spectra of HULIS from fog water are reported to have peaks*

*at shorter excitation and emission wavelengths than those of terrestrial fulvic acids (Graber and Rudich, 2006).' – This sentence seems out of place here. What does it contribute to the discussion here?*
**Reply:** Done.

*l. 549: Can you quantify 'a few' in this context?*
**Reply:** It refers to small organic acids, and it was changed to "organic acids with a few carbon atoms" according to your suggestion in last time. Since we did not perform measurements of organic acids here, it is difficult to know their structures. But traditionally, they may refers to C1-C6 organic acids.

*l. 579: This still adds to the confusion I had pointed out in the last round of reviews: Are you implying that HULIS cannot be high MW oligomers?*
**Reply:** Very sorry if it is still not clear to you. What we want to say is that since high MW oligomers can be formed in the initial stage of oxidation, while HULIS concentration also quickly increase in the initial stage of oxidation as we observed here, so we imply that HULIS can be (*not cannot be*) some of the high MW oligomers. This is made clear in the text.

*l. 610: 'simulated lights' is not correct. Do you mean 'light intensity' or 'wavelength' – or both?*
**Reply:** should be both.

*l. 641/2: 'As a result, all data points located outside the f44 vs. f43 space established by Ng et al. (2010) for ambient aerosols, owing to the relatively low f43 values.'*
*This sentence seems grammatically wrong; it is also not clear what you try to say.*
**Reply:** It is now changed to, "As a result, all data points located outside the established $f_{44}$ vs. $f_{43}$ region (bounded by the two dash lines in Figs. 9d-f) by Ng et al. (2010) for ambient aerosols."

*l. 725: Can you quantify the fraction of HULIS to total reaction products?*
**Reply:** It is practically difficult to do so in this work.

*l. 750: 'This finding further indicates the effectiveness of DTT method to represent OP.' Which finding do you refer to here? Isn't this a circular and redundant statement? – The DTT method was developed to quantify OP – thus, it seems obvious that it can used to do this.*
**Reply:** This sentence is deleted.

*l. 791: replace 'photolyze itself' by 'undergo direct photolysis'*
*l. 800: replace 'foil' by 'fuel'*
**Reply:** Done

*l. 801 – 803: 'Aqueous oxidation of 4-nitrophenol with OH can lead to a photobleaching effect too.' Nitrophenols are not comparable at all to the compounds you studied. There, the nitro group causes light absorption – thus this comparison seems out pf place and not relevant.*
**Reply:** Agree. This sentence is deleted.

*Data availability: Please deposit your data in a suitable repository according to the journal data policy https://www.atmospheric-chemistry-and-physics.net/policies/data_policy.html*
**Reply:** As suggested, we have deposit the data at : http://nuistairquality.com/eugenol_data_and_figure

---

## Author Response (AR5)

*Response to comments*

**Dear Editor**

**Thank you very much again for your comments regarding our manuscript! We have addressed your comments (repeating in *italics*) below. The manuscript was also proofread again, please check the manuscript with tracked changes.**

*l. 24: What is the product of the reaction of HO2 and eugenol? Is molecular oxygen formed or H2O2? If the former, wouldn't it be then a reduction reaction? You could replace 'oxidize' by 'degrade'.*

**Reply:** Thanks, it should be molecular oxygen, and the word "oxidize" is replaced by "degrade"

*Figures 1, 4, 7, 8, 9, 10: Please change the figure according to the legend., i.e. replacing 'Direct', 'OH', and 3C\*' by 'direct photolysis, OH-initiated oxidation and 3C\*-initiated oxidation, respectively.*

**Reply:** As suggested, the legends as well as those in supplement have been changed.

*l. 569: replace 'contributed' by 'contribute'*
*l. 614: replace 'dramatic' by 'significant' or 'rapid' – depending on what matches the content*
*l. 631: remove 'the behaviors of'*
*l. 662: replace 'dominates' by 'dominate'*
l. 694: Please write consistently 'OH', not HO· (check the remainder of the manuscript)
l. 787: replace 'foil' by 'fuel'

**Reply:** Done